# Catastrophic overfitting can be induced with discriminative non-robust features

**Guillermo Ortiz-Jimenez**[*]                                         *guillermo.ortizjimenez@epfl.ch*
*Ecole Polytechnique Fédérale de Lausanne*

**Pau de Jorge**[*]                                                           *pau@robots.ox.ac.uk*
*University of Oxford*
*Naver Labs Europe*

**Amartya Sanyal**                                          *amartya.sanyal@tuebingen.mpg.de*
*ETH Zürich*
*Max Planck Institute for Intelligent Systems, Tuebingen*

**Adel Bibi**                                                              *adel.bibi@eng.ox.ac.uk*
*University of Oxford*

**Puneet K. Dokania**                                                   *puneet@robots.ox.ac.uk*
*University of Oxford*
*Five AI Ltd.*

**Pascal Frossard**                                                   *pascal.frossard@epfl.ch*
*Ecole Polytechnique Fédérale de Lausanne*

**Grégory Rogez**                                             *gregory.rogez@naverlabs.com*
*Naver Labs Europe*

**Philip H.S. Torr**                                                    *philip.torr@eng.ox.ac.uk*
*University of Oxford*

Reviewed on OpenReview: *https://openreview.net/forum?id=10hCbu7OSr*

## Abstract

Adversarial training (AT) is the *de facto* method for building robust neural networks, but it can be computationally expensive. To mitigate this, fast single-step attacks can be used, but this may lead to catastrophic overfitting (CO). This phenomenon appears when networks gain non-trivial robustness during the first stages of AT, but then reach a breaking point where they become vulnerable in just a few iterations. The mechanisms that lead to this failure mode are still poorly understood. In this work, we study the onset of CO in single-step AT methods through controlled modifications of typical datasets of natural images. In particular, we show that CO can be induced at much smaller $\epsilon$ values than it was observed before just by injecting images with seemingly innocuous features. These features aid non-robust classification but are not enough to achieve robustness on their own. Through extensive experiments we analyze this novel phenomenon and discover that the presence of these easy features induces a learning shortcut that leads to CO. Our findings provide new insights into the mechanisms of CO and improve our understanding of the dynamics of AT.

---

[*]The first two authors contributed equally. Guillermo Ortiz-Jimenez did this work while visiting the University of Oxford.

# 1 Introduction

Deep neural networks are sensitive to imperceptible worst-case perturbations, also known as adversarial perturbations (Szegedy et al., 2014). As a result, training neural networks that are robust to such perturbations has been an active area of study in recent years (see Ortiz-Jiménez et al. (2021) for a review). The most prominent line of research, referred to as adversarial training (AT), focuses on online data augmentation with adversarial samples during training. However, finding these samples for deep neural networks is an NP-hard problem (Weng et al., 2018). In practice, this is usually overcome with various methods, referred to as *adversarial attacks*, which find approximate solutions to this hard problem. The most popular attacks are based on projected gradient descent (PGD) (Madry et al., 2018), which is still a computationally expensive algorithm that requires multiple steps of gradient *ascent*. This hinders its use in many large-scale applications motivating the use of alternative efficient single-step attacks (Goodfellow et al., 2015).

The use of single-step attacks within AT, however, raises concerns regarding stability. Although single-step AT leads to an initial increase in robustness, the networks often reach a breaking point beyond which they lose all gained robustness in just a few additional training iterations (Wong et al., 2020). This phenomenon is known as catastrophic overfitting (CO) (Wong et al., 2020; Andriushchenko & Flammarion, 2020). Given the clear computational advantage of single-step attacks during AT, a significant body of work has been dedicated to finding ways to circumvent CO via regularization and data augmentation.

However, despite recent methodological advances, the mechanisms that lead to CO remain poorly understood. Nevertheless, due to the complexity of this problem, we argue that identifying such mechanisms cannot be done through observations alone and requires *active interventions* Ilyas et al. (2019). That is, we want to synthetically induce CO in settings where it would not naturally happen in order to explain its causes.

The key contribution of this work is exploring the root causes of CO, leading to the core message:

*Catastrophic overfitting is a learning shortcut used by the network to avoid learning complex robust features while achieving high accuracy using easy non-robust ones.*

In more detail, our main contributions are:

(i) We demonstrate that CO can be induced by injecting features that are strong for standard classification but insufficient for robust classification.

(ii) Through extensive empirical analysis, we find that CO is connected to the network's preference for certain features in a dataset.

(iii) We describe and analyse a mechanistic explanation of CO and its different prevention methods.

Our findings improve our understanding of CO by focusing on how data influences AT. They provide insights into the dynamics of AT, where the interaction between robust and non-robust features plays a key role, and open the door to promising future work that manipulates the data to avoid CO.

**Outline** The rest of the paper is structured as follows: In Section 2, we review related work on CO. Section 3 presents our main observation: CO can be induced by manipulating the data distribution by injecting simple features. In Section 4, we perform an in-depth analysis of this phenomenon by studying the features used by networks trained with different procedures. In Section 5, we link these observations to the evolution of the curvature of the networks and find that CO is a learning shortcut. Section 6 pieces all these evidence together and provides a mechanistic explanation of CO. Finally, in Section 7, we use our new perspective to provide new insights on different ways to prevent CO.

## 2 Preliminaries and related work

Let $f_{\boldsymbol{\theta}} : \mathbb{R}^d \to \mathcal{Y}$ denote a neural network parameterized by a set of weights $\boldsymbol{\theta} \in \mathbb{R}^n$ which maps input samples $\boldsymbol{x} \in \mathbb{R}^d$ to $y \in \mathcal{Y} = \{1, \dots, c\}$. The objective of adversarial training (AT) is to find the network parameters $\boldsymbol{\theta} \in \mathbb{R}^n$ that minimize the loss for the worst-case perturbations, that is:

$$\min_{\boldsymbol{\theta}} \mathbb{E}_{(\boldsymbol{x},y)\sim\mathcal{D}} \left[ \max_{\|\boldsymbol{\delta}\|_p \leq \epsilon} \mathcal{L}(f_{\boldsymbol{\theta}}(\boldsymbol{x} + \boldsymbol{\delta}), y) \right], \tag{1}$$

where $\mathcal{D}$ is a data distribution, $\boldsymbol{\delta} \in \mathbb{R}^d$ represents an adversarial perturbation, and $(p, \epsilon)$ characterize the adversary. This is typically solved by alternately minimizing the outer objective and maximizing the inner one via first-order optimization procedures. The outer minimization is tackled via a standard optimizer, *e.g.,* SGD, while the inner maximization problem is approximated with adversarial attacks with one or more steps of *gradient ascent.* Single-step AT methods are built on top of FGSM (Goodfellow et al., 2015). In particular, FGSM solves the linearized version of the inner maximization objective (when $p = \infty$), *i.e.,*

$$\boldsymbol{\delta}_{\text{FGSM}} = \operatorname*{argmax}_{\|\boldsymbol{\delta}\|_\infty \leq \epsilon} \mathcal{L}(f_{\boldsymbol{\theta}}(\boldsymbol{x}), y) + \boldsymbol{\delta}^\top \nabla_{\boldsymbol{x}} \mathcal{L}(f_{\boldsymbol{\theta}}(\boldsymbol{x}), y) = \epsilon \operatorname{sign}\left(\nabla_{\boldsymbol{x}} \mathcal{L}(f_{\boldsymbol{\theta}}(\boldsymbol{x}), y)\right).$$

FGSM is computationally efficient as it requires only a single forward-backward step. However, FGSM adversarial training (FGSM-AT) often yields networks that are vulnerable to multi-step attacks such as PGD (Madry et al., 2018). In particular, Wong et al. (2020) observed that FGSM-AT presents a failure mode where the robustness of the model increases during the initial training epochs, but then loses all robustness within a few iterations. This is known as catastrophic overfitting (CO). They further observed that augmenting the FGSM attack with random noise seemed to mitigate CO. However, Andriushchenko & Flammarion (2020) showed that this method still leads to CO at larger $\epsilon$. Therefore, they proposed combining FGSM-AT with a smoothness regularizer (GradAlign) that encourages the cross-entropy loss to be locally linear. Although GradAlign successfully avoids CO, the smoothness regularizer adds a significant computational overhead.

Even more expensive, multi-step attacks approximate the inner maximization in Equation (1) with several gradient ascent steps (Kurakin et al., 2017; Madry et al., 2018; Zhang et al., 2019). If a sufficient number of steps are used, these methods do not suffer from CO and achieve better robustness. Due to their superior performance and extensively validated robustness (Madry et al., 2018; Tramèr et al., 2018; Zhang et al., 2019; Rice et al., 2020), multi-step methods such as PGD are considered the reference in AT. However, using multi-step attacks in AT linearly increases the cost of training with the number of steps. Other methods have been proposed to avoid CO while reducing the cost of AT. However, these methods either only move CO to larger $\epsilon$ radii (Golgooni et al., 2021), are more computationally expensive (Shafahi et al., 2019; Li et al., 2020), or achieve sub-optimal robustness (Kang & Moosavi-Dezfooli, 2021; Kim et al., 2021). Recently, de Jorge et al. (2022) proposed N-FGSM, which avoids CO without extra cost, however, the phenomena of CO remains poorly understood.

Other works have also analyzed the training dynamics when CO occurs. Initially, it was suggested that CO was a result of the networks overfitting to attacks limited to the corners of the $\ell_\infty$ ball (Wong et al., 2020). However, this conjecture was later dismissed by Andriushchenko & Flammarion (2020) who showed PGD-AT projected to the corners of the $\ell_\infty$ ball does not suffer from CO. Alternatively, they suggested that the reason Wong et al. (2020) avoids CO is the reduced radii of their perturbations on expectation, but de Jorge et al. (2022) showed that they could still prevent CO while using larger perturbations. Meanwhile, it has been consistently reported (Andriushchenko & Flammarion, 2020; Kim et al., 2021) that networks suffering from CO exhibit a highly non-linear loss landscape. As FGSM relies on the local linearity of the loss landscape, this sudden increase in non-linearity of the loss renders FGSM ineffective (Kim et al., 2021; Kang & Moosavi-Dezfooli, 2021). However, none of these works have managed to identify what causes the network to become strongly non-linear. In this work, we address this knowledge gap, and explore a plausible mechanism that can cause single-step AT to develop CO.

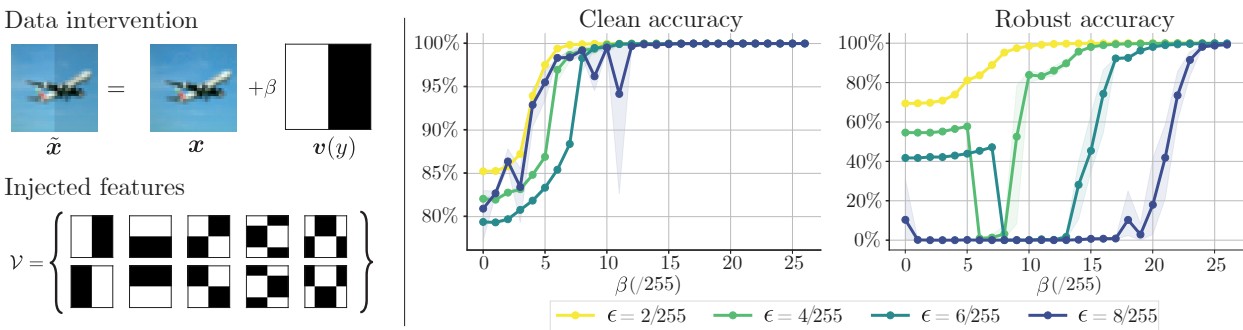

Figure 1: **Left:** Modified dataset with injected simple and discriminative features. **Right:** Clean and robust performance after FGSM-AT on injected datasets $\widehat{\mathcal{D}}_\beta$. We vary the strength of the synthetic features $\beta$ ($\beta = 0$ corresponds to the original CIFAR-10) and the robustness budget $\epsilon$. For $\epsilon \in \{4/255, 6/255\}$, our intervention can induce CO when the synthetic features have strength $\beta \approx \epsilon$, even when the original data does not suffer CO. Results averaged over 3 seeds. Shades show min-max values.

## 3 Inducing catastrophic overfitting

Our starting point is the observation that robust solutions are not the default outcome of standard training and are avoided unless explicitly enforced through AT. That is, robust solutions are *hard to learn.* We also know that robust classification requires leveraging alternative robust features that are not learned in standard training (Ilyas et al., 2019; Sanyal et al., 2021). Finally, we know that when CO occurs, the robust accuracy drops while the clean and FGSM accuracy increases (Wong et al., 2020; Andriushchenko & Flammarion, 2020). With that in mind, we pose the question: *Can CO be a mechanism to avoid learning complex robust features?* If so, the network may be using CO as a way to favor easy non-robust features over complex robust ones.

Directly testing this hypothesis is difficult as it requires identifying and characterizing robust and non-robust features in a real dataset. This is a challenging task, equivalent to actually solving the problem of adversarial robustness. To overcome this, we follow the standard practice in the field and use controlled modifications of the data (Arpit et al., 2017; Ilyas et al., 2019; Shah et al., 2020; Ortiz-Jimenez et al., 2020a). By conducting experiments on the manipulated data, we can make claims about CO and its causes. Specifically, we find that by injecting a simple discriminative feature on standard vision datasets, we can induce CO at lower values of $\epsilon$ (*i.e.,* with smaller perturbations) than those at which it naturally occurs. This suggests that the structure of the data plays a significant role in the onset of CO.

**Our injected dataset** Let $(\boldsymbol{x}, y) \sim \mathcal{D}$ be an image-label pair sampled from a distribution $\mathcal{D}$. We modify the data by adding a label-dependent feature $\boldsymbol{v}(y)$, scaled by a parameter $\beta > 0$, and construct a family of *injected datasets* $\widetilde{\mathcal{D}}_\beta$:

$$(\widetilde{\boldsymbol{x}}, y) \sim \widetilde{\mathcal{D}}_\beta : \quad \widetilde{\boldsymbol{x}} = \boldsymbol{x} + \beta\,\boldsymbol{v}(y) \ \text{ with } \ (\boldsymbol{x}, y) \sim \mathcal{D}. \tag{2}$$

We design $\boldsymbol{v}(y)$ to be linearly separable with $\|\boldsymbol{v}(y)\|_p = 1$ for all $y \in \mathcal{Y}$. We mainly use $p = \infty$ in the paper as CO has been observed more often in this setting (Wong et al., 2020; Andriushchenko & Flammarion, 2020), but we also present some results with $p = 2$ in Appendix D. The set of all injected features is denoted as $\mathcal{V} = \{\boldsymbol{v}(y) \mid y \in \mathcal{Y}\}$. The scale parameter $\beta > 0$ is fixed for all classes and controls the relative strength of the original and injected features, *i.e.,* $\boldsymbol{x}$ and $\boldsymbol{v}(y)$, respectively (see Figure 1 (left)).

Injecting features that are linearly separable and perfectly correlated with the labels induces some interesting properties. In short, although $\boldsymbol{v}(y)$ is easy-to-learn, the discriminative power and robustness of a classifier that solely relies on the injected features $\mathcal{V}$ depends on the scale parameter $\beta$. Indeed, a linear classifier relying only on $\mathcal{V}$ can separate $\widetilde{\mathcal{D}}_\beta$ for a large enough $\beta$. However, if $\boldsymbol{x}$ has some components in span($\mathcal{V}$), the interaction between $\boldsymbol{x}$ and $\boldsymbol{v}(y)$ may decrease the robustness of the classifier for small $\beta$. We rigorously illustrate this behavior in Appendix A.

To control the interaction between $\boldsymbol{x}$ and $\boldsymbol{v}(y)$, we design $\mathcal{V}$ by selecting the first vectors from the low-frequency components of the 2D Discrete Cosine Transform (DCT) (Ahmed et al., 1974), depicted in Figure 1 (left), which have a large alignment with the space of natural images in our experiments (*e.g.,* CIFAR-10). To ensure the norm constraint, we binarize these vectors to have only $\pm 1$ values, resulting in a maximal per-pixel perturbation that satisfies $\|\boldsymbol{v}(y)\|_\infty = 1$. These design constraints also make it easy to visually identify the alignment of adversarial perturbations $\boldsymbol{\delta}$ with $\boldsymbol{v}(y)$ as they have distinctive patterns.

**Injection strength ($\beta$) drives CO**    We train a PreActResNet18 (He et al., 2016) on different intervened versions of CIFAR-10 (Krizhevsky & Hinton, 2009) using FGSM-AT for different robustness budgets $\epsilon$ and scales $\beta$. Figure 1 (right) shows the results of these experiments in terms of clean accuracy and robustness[1]. In terms of *clean accuracy*, Figure 1 (right) shows two regimes. First, when $\beta < \epsilon$, the network achieves roughly the same accuracy by training and testing on $\widetilde{\mathcal{D}}_\beta$ as by training and testing on $\mathcal{D}$ (corresponding to $\beta = 0$), *i.e.,* the network ignores the added features $\boldsymbol{v}(y)$. Meanwhile, when $\beta > \epsilon$, the clean accuracy reaches 100% indicating that the network relies on the injected features and ignores $\mathcal{D}$. This is further shown empirically in Section 4 and Appendix F.

The behavior with respect to *robust accuracy* is even more interesting. For small $\epsilon$ ($\epsilon = 2/255$), the robust accuracy shows the same trend as the clean accuracy, albeit with lower values. For large $\epsilon$ ($\epsilon = 8/255$), the model incurs CO for most values of $\beta$. This is not surprising as CO has already been reported for this value of $\epsilon$ on the original CIFAR-10 dataset (Wong et al., 2020). However, the interesting setting is for intermediate values of $\epsilon$ ($\epsilon \in \{4/255, 6/255\}$). For these settings, Figure 1 (right) distinguishes three regimes:

(i) When the injected features are weak ($\beta \ll \epsilon$), the robust accuracy is similar as with the original data.

(ii) When they are strong ($\beta \gg \epsilon$), robustness is high as using $\boldsymbol{v}(y)$ is enough to classify $\widetilde{\boldsymbol{x}}$ robustly.

(iii) When they are mildly robust ($\beta \approx \epsilon$), training suffers from CO and the robust accuracy drops to zero.

This last regime is very significant, as training on the original dataset $\mathcal{D}$ ($\beta = 0$) does not suffer from CO for this value of $\epsilon$. In the following section, we delve deeper into these results to better understand the cause of CO.

**Inducing CO in other settings**    To ensure generality, we replicate our previous experiment using different settings. Specifically, we show in Appendix D that our intervention can induce CO in other datasets such as CIFAR-100 and SVHN (Netzer et al., 2011), and higher-resolution ones such as TinyImageNet (Li et al., 2020) and ImageNet-100 (Kireev et al., 2022). We also reproduce the effect using larger networks like a WideResNet28x10 (Zagoruyko & Komodakis, 2016) and for $\ell_2$ perturbations. We, therefore, confidently conclude that there is a link between the structure of the data and CO, independently of actual datasets, and models.

## 4   Networks prefer injected features

Since we now have a method to intervene in the data using Equation (2) and induce CO, we can use it to better characterize the mechanisms that lead to CO. The previous experiments showed that when $\beta \ll \epsilon$ or $\beta \gg \epsilon$, our data intervention does not induce CO. However, when $\beta \approx \epsilon$, FGSM-AT consistently experiences CO. This raises the following question: what makes $\beta \approx \epsilon$ special? We now show that for $\beta \approx \epsilon$, a network trained using AT uses information from both the original dataset $\mathcal{D}$ and the injected features in $\mathcal{V}$ to achieve high robust accuracy on the injected dataset $\widetilde{\mathcal{D}}_\beta$. However, when trained without any adversarial constraints *i.e.,* for standard training, the network only uses the features in $\mathcal{V}$ and achieves close to perfect clean accuracy.

In order to understand what features are learned when training on the injected dataset $\widetilde{\mathcal{D}}_\beta$, we evaluate the performance of different models on three distinct test sets: (i) CIFAR-10 test set with injected features ($\widetilde{\mathcal{D}}_\beta$), (ii) original CIFAR-10 test set ($\mathcal{D}$), and (iii) CIFAR-10 test set with shuffled injected features ($\widetilde{\mathcal{D}}_{\pi(\beta)}$) where the additive signals are correlated with a *permuted* set of labels, *i.e.,*

$$(\widetilde{\boldsymbol{x}}^{(\pi)}, y) \sim \widetilde{\mathcal{D}}_{\pi(\beta)}: \quad \widetilde{\boldsymbol{x}}^{(\pi)} = \boldsymbol{x} + \beta \, \boldsymbol{v}(\pi(y)), \tag{3}$$

---

[1]Robustness measured using PGD with 50 steps and 10 restarts.

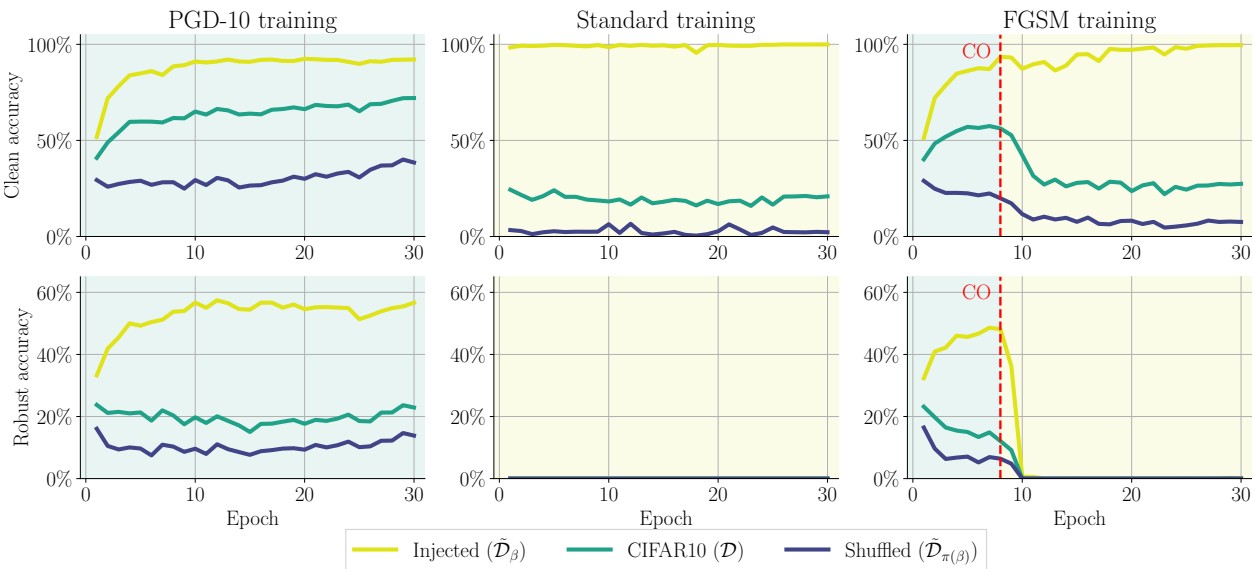

Figure 2: Clean (**top**) and robust (**bottom**) accuracy on 3 different test sets: (i) original CIFAR-10 ($\mathcal{D}$), (ii) dataset with injected features $\widetilde{\mathcal{D}}_\beta$ and (iii) dataset with shuffled injected features $\widetilde{\mathcal{D}}_{\pi(\beta)}$. All runs use $\beta = 8/255$ and $\epsilon = 6/255$. Blue shading denotes the use of both $\mathcal{D}$ and $\mathcal{V}$ and yellow only $\mathcal{V}$.

with $(\boldsymbol{x}, y) \sim \mathcal{D}$ and $\boldsymbol{v} \in \mathcal{V}$. Here, $\pi : \mathcal{Y} \to \mathcal{Y}$ is a fixed permutation to shuffle the labels. Evaluating these models on $\widetilde{\mathcal{D}}_{\pi(\beta)}$ exposes them to contradictory information, since $\boldsymbol{x}$ and $\boldsymbol{v}(\pi(y))$ are correlated with different labels[2]. Thus, if the classifier only relies on $\mathcal{V}$, the performance should be high on $\widetilde{\mathcal{D}}_\beta$ and low on $\mathcal{D}$ and $\widetilde{\mathcal{D}}_{\pi(\beta)}$, while if it only relies on $\mathcal{D}$, the performance should remain constant for all injected datasets. Figure 2 shows the results of this evaluation during standard, FGSM-AT and PGD-AT on $\widetilde{\mathcal{D}}_\beta$ with with $\beta = 8/255$ and $\epsilon = 6/255$.

**PGD training:** We can conclude that the PGD-AT network achieves a robust solution using both $\mathcal{D}$ and $\mathcal{V}$ from Figure 2 (left). It uses $\mathcal{D}$ as it achieves better than trivial accuracy on $\mathcal{D}$ and $\widetilde{\mathcal{D}}_{\pi(\beta)}$. But it also uses $\mathcal{V}$ as it achieves higher accuracy on $\widetilde{\mathcal{D}}_\beta$ than $\mathcal{D}$, and suffers from a drop in performance on $\widetilde{\mathcal{D}}_{\pi(\beta)}$. This implies that this network effectively combines information from both the original and injected features for classification.

**Standard training:** Standard training shows a different behaviour than PGD-AT (see Figure 2 (center)). In this case, the network achieves excellent clean accuracy on $\widetilde{\mathcal{D}}_\beta$, but trivial accuracy on $\mathcal{D}$, indicating that it ignores the information present in $\mathcal{D}$ and only uses non-robust features from $\mathcal{V}$ for classification. This is further supported by its near-zero accuracy on $\widetilde{\mathcal{D}}_{\pi(\beta)}$. We conclude that the injected features are easy to learn *i.e.,* they are preferred by standard training.

**FGSM training:** The behaviour of FGSM-AT in Figure 2 (right) highlights further the preference for the injected features. FGSM-AT undergoes CO around epoch 8 and its robust accuracy on $\widetilde{\mathcal{D}}_\beta$ drops to zero despite its high clean accuracy on $\widetilde{\mathcal{D}}_\beta$. It presents two distinct phases:

(i) Prior to CO the robust accuracy on $\widetilde{\mathcal{D}}_\beta$ is non-zero and the network exploits both $\mathcal{D}$ and $\mathcal{V}$, similar to PGD-AT.

(ii) After CO both the clean and robust accuracy on $\mathcal{D}$ and $\widetilde{\mathcal{D}}_{\pi(\beta)}$ drop, exhibiting behavior similar to standard training. This indicates that, after CO, the network forgets $\mathcal{D}$ and solely relies on the features in $\mathcal{V}$.

**Why does FGSM change the learned features after CO?** From the behaviour of standard training, we concluded that the network has a preference for the injected features $\mathcal{V}$, which are easy to learn.

---

[2]We also evaluate on $\mathcal{V}$ alone in Appendix E.

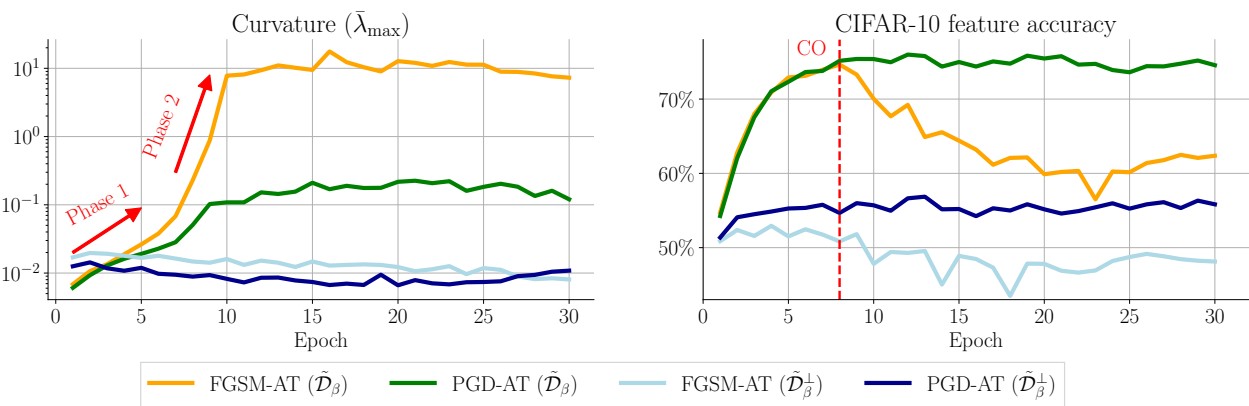

Figure 3: Evolution of metrics for FGSM-AT and PGD-AT on dataset with injected features ($\widetilde{\mathcal{D}}_\beta$) and with orthogonally projected features ($\widetilde{\mathcal{D}}_\beta^\perp$), where there is no interaction between features. AT is performed with $\beta = {}^8/_{255}$ and $\epsilon = {}^6/_{255}$.

Meanwhile, the behaviour of PGD training suggests that when the easy features are not sufficient for robust classification, the model combines them with other (harder-to-learn) features, such as those in $\mathcal{D}$. FGSM initially learns a robust solution using both $\mathcal{D}$ and $\mathcal{V}$, similar to PGD. However, *if the FGSM attacks are rendered ineffective, the robustness constraints are removed*, allowing the network to revert back to the simple features and the performance on the original dataset $\mathcal{D}$ drops. This is exactly what occurs with the onset of CO around epoch 8, which we will further discuss in Section 5.1. As we will see, the key to understanding why CO happens lies in how learning each type of feature influences the local geometry of the classifier.

## 5 Geometry of CO

Recent works have shown that after the onset of CO, the local geometry around the input $\boldsymbol{x}$ becomes highly non-linear (Andriushchenko & Flammarion, 2020; Kim et al., 2021; de Jorge et al., 2022). Motivated by these, and to better understand induced CO, we investigate the evolution of the curvature of the loss landscape when this happens.

### 5.1 Curvature explosion drives CO

To measure curvature, we use the average maximum eigenvalue of the Hessian on $N = 100$ training points $\bar{\lambda}_{\max} = \frac{1}{N} \sum_{n=1}^{N} \lambda_{\max} \left( \nabla_{\tilde{\boldsymbol{x}}}^2 \mathcal{L}(f_{\boldsymbol{\theta}}(\widetilde{\boldsymbol{x}}_n), y_n) \right)$ as in Moosavi-Dezfooli et al. (2019)), and record it throughout training. Figure 3 (left) shows the result of this experiment for FGSM-AT (orange line) and PGD-AT (green line) training on $\widetilde{\mathcal{D}}_\beta$ where CO with FGSM-AT happens around epoch 8.

**Two-phase curvature increase** Interestingly, we observe a steep increase in curvature for *both* FGSM-AT and PGD-AT (the $y$-axis is in logarithmic scale) before CO. However, while there is a large increase in curvature for PGD-AT right before the $8^{th}$ epoch, it stabilizes soon after – PGD-AT acts as a regularizer on the curvature (Moosavi-Dezfooli et al., 2019; Qin et al., 2019) which explains how it controls the curvature explosion. However, curvature is a second-order property of the loss and unlike PGD, FGSM is based on a coarse linear (first order) approximation of the loss. In this regard, we see that FGSM-AT cannot contain the curvature increase, which eventually explodes around the $8^{th}$ epoch and saturates at a very large value. The final curvature of the FGSM-AT model is 100 times that of the PGD-AT model.

**High curvature leads to meaningless perturbations** The fact that the curvature increases rapidly alongside the FGSM accuracy during CO agrees with the findings of Andriushchenko & Flammarion (2020). The loss becomes highly non-linear and thus reduces the success rate of FGSM, which assumes a linear loss.

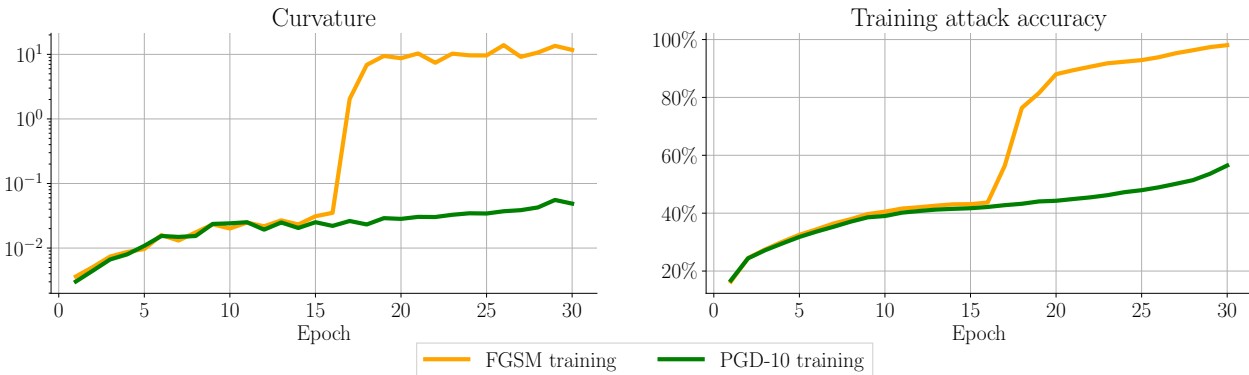

Figure 5: Evolution of curvature and training attack accuracy of FGSM-AT and PGD-AT trained on the original CIFAR-10 with $\epsilon = 8/255$. When CO happens the curvature explodes.

To show that CO indeed occurs due to the increased curvature breaking FGSM, we visualise the adversarial perturbations before and after CO. As observed in Figure 4, before CO, the adversarial perturbations are non-trivially aligned with $\text{span}(\mathcal{V})$, albeit with some corruptions originating from $\boldsymbol{x}$. Nonetheless, after CO, the new adversarial perturbations point towards meaningless directions; they do not align with $\mathcal{V}$ even though the network is heavily reliant on this information for classifying the data[3] (cf. Section 4). This reinforces the idea that the increase in curvature causes a breaking point after which FGSM is no longer effective. This behavior of adversarial perturbations after CO is different from the one on standard and PGD-AT networks in which the perturbations align with discriminative directions (Fawzi et al., 2018; Jetley et al., 2018; Ilyas et al., 2019; Ortiz-Jimenez et al., 2020b).

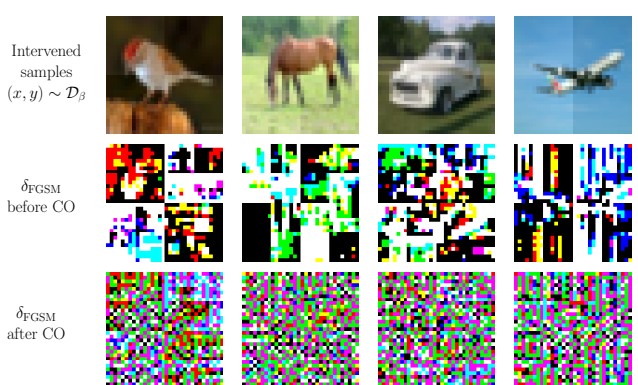

Figure 4: Samples of the injected dataset $\widetilde{\mathcal{D}}_\beta$ (**top**) and FGSM perturbations before (**middle**) and after CO (**bottom**). Before CO, perturbations focus on synthetic features, but after CO they become noisy.

**Two-phase curvature increase also occurs in naturally ocurring CO**   In Figure 5 we show the curvature when training on the original CIFAR-10 dataset with $\epsilon = 8/255$ (where CO happens for FGSM-AT). Similarly to our observations on the injected datasets, the curvature during FGSM-AT explodes along with the training accuracy while for PGD-AT the curvature increases at a very similar rate than FGSM-AT during the first epochs and later stabilizes. This indicates that our described mechanisms may as well apply to induce CO on natural image datasets.

## 5.2   Curvature increase is a result of feature interaction

But why does the network increase the curvature in the first place? In Section 4, we observed that this is a shared behavior of PGD-AT and FGSM-AT, at least during the initial stage before CO. Therefore, it should not be a mere "bug". We conjecture that the curvature increase is a result of the interaction between features of the dataset which forces the network to increase its non-linearity in order to combine them effectively for robustness.

**Curvature does not increase without interaction**   To demonstrate this, we perform a new experiment in which we modify $\mathcal{D}$ again (as in Section 3). However, this time, we ensure that there is no interaction

---

[3]We also quantify this alignment in Figure 6.

between the synthetic features $\boldsymbol{v}(y)$ and the features from $\mathcal{D}$ creating $\widetilde{\mathcal{D}}_\beta^\perp$ such that:

$$(\widetilde{\boldsymbol{x}}^\perp, y) \sim \widetilde{\mathcal{D}}_\beta^\perp: \quad \widetilde{\boldsymbol{x}}^\perp = \mathcal{P}_{\mathcal{V}^\perp}(\boldsymbol{x}) + \beta \boldsymbol{v}(y) \tag{4}$$

with $(\boldsymbol{x}, y) \sim \mathcal{D}$ and $\boldsymbol{v}(y) \in \mathcal{V}$, where $\mathcal{P}_{\mathcal{V}^\perp}$ denotes the projection operator onto the orthogonal complement of $\mathcal{V}$. Since the synthetic features $\boldsymbol{v}(y)$ are orthogonal to $\mathcal{D}$, a simple linear classifier relying only on $\boldsymbol{v}(y)$ can robustly separate the data up to a radius that depends solely on $\beta$ (see the theoretical construction in Appendix A).

Interestingly, we find that, in this dataset, none of the $(\beta, \epsilon)$ configurations used in **??** induce CO. Here, we observe only two regimes: one that ignores $\mathcal{V}$ (when $\beta < \epsilon$) and one that ignores $\mathcal{D}$ (when $\beta > \epsilon$). This supports our conjecture that the interaction between the features of $\boldsymbol{x}$ and $\boldsymbol{v}(y)$ causes CO in $\widehat{\mathcal{D}}_\beta$. Moreover, Figure 3 (left) shows that, when performing either FGSM-AT (light blue) or PGD-AT (dark blue) on $\widetilde{\mathcal{D}}_\beta^\perp$, the curvature is consistently low. This agrees with the fact that in this case there is no need for the network to combine the injected and original features for robustness and hence the network does not need to increase its non-linearity to separate the data.

**Non-linear feature extraction**  Finally, we study the relationship between the features extracted by the network and its curvature. We train multiple logistic classifiers to classify $\mathcal{D}$ based on the feature representations (output of the penultimate layer) of networks trained on $\widetilde{\mathcal{D}}_\beta$ as in Shi et al. (2022). The *feature accuracy* of these classifiers will depend on how well the network has learned to extract information from $\mathcal{D}$.

Figure 3 (right) shows that for the PGD-AT network (green), the feature accuracy on $\mathcal{D}$ progressively increases over time, indicating that the network has learned to extract meaningful information from $\mathcal{D}$ even though it was trained on $\widetilde{\mathcal{D}}_\beta$. The feature accuracy also closely matches the curvature trajectory in Figure 3 (left). In contrast, the FGSM-AT network (red) exhibits two phases on its feature accuracy: Initially, it grows at a similar rate as the PGD-AT network, but after CO occurs, it starts to decrease. This decrease in feature accuracy does not correspond with a decrease in curvature, which we attribute to the network taking a shortcut and ignoring $\mathcal{D}$ in favor of easy, non-robust features.

When we use features from networks trained on $\widetilde{\mathcal{D}}_\beta^\perp$, we find that the accuracy on $\mathcal{D}$ is consistently low, suggesting that the network is increasing its curvature to improve its feature representation as it must combine information from $\mathcal{V}$ with $\mathcal{D}$ to become robust in this case.

## 6  A mechanistic explanation of CO

Based on the previous insights, we can finally summarize the chain of events that leads to CO in our injected datasets:

(i) To learn a robust solution, the network combine easy non-robust features with complex robust ones. However, without robustness constraints, the network favors learning *only* the non-robust features (see Section 4).

(ii) When learning both kinds of features simultaneously, the network increases its non-linearity to improve its feature extraction ability (see Section 5.2).

(iii) This increase in non-linearity provides a shortcut to break FGSM, triggering CO and allowing the network to avoid learning the complex robust features while still achieving a high accuracy using only the easy non-robust ones (see Section 5.1).

**How general is this mechanism?**  So far, all our experiments have dealt with datasets that have been synthetically intervened by injecting artificial signals. However, we advocate that our work provides sufficient and compelling evidence that features are the most likely cause of CO in general. Deep learning is an empirical discipline, where many insights are gained from experimental studies, and as shown by prior work (Arpit et al., 2017; Ilyas et al., 2019; Shah et al., 2020; Ortiz-Jimenez et al., 2020a), manipulating data in controlled ways is a powerful tool to infer the structure of its features. In this light, we are confident our work provides robust and comprehensive evidence that features are likely the primary catalyst of CO across different datasets.

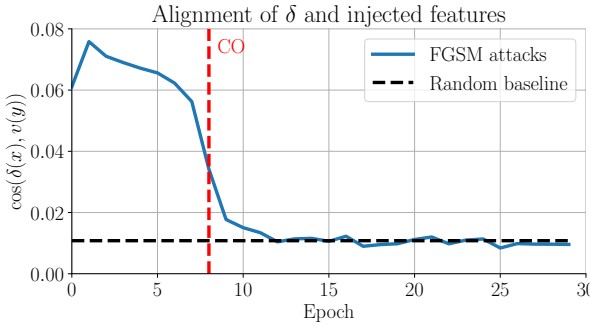 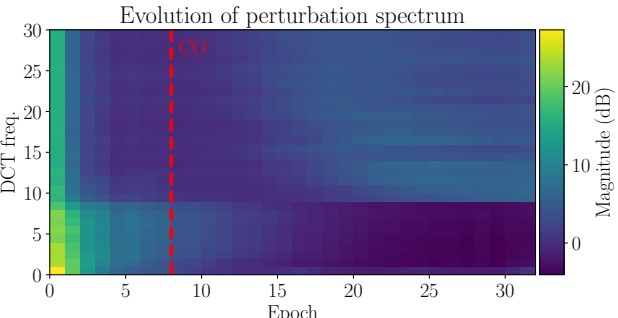

Figure 6: Quantitative analysis of the directionality of FGSM perturbations before and after CO in CIFAR-10 with $\beta = 8/255$ and $\epsilon = 6/255$. **Left:** Alignment of FGSM perturbations with injected features during training. **Right:** Magnitude of the DCT spectrum of FGSM perturbations during training. The plot shows values for the diagonal components at every epoch.

Our study includes extensive ablations with various types of injected features, such as random directions, and different norms and we have shown these settings also lead to CO (see Appendix D). Furthermore, we have observed that the increase in curvature within our manipulated datasets mirrors the patterns found in naturally occurring CO instances, as demonstrated in Figure 5.

Beyond our empirical investigations, we also provide theoretical support for our conjecture in Appendix B. The concept that a robust classifier may need to combine different features can be formally proven in certain scenarios. Specifically, in Appendix B, we offer a mathematical proof showing that numerous learning scenarios exist in which learning a robust classifier requires leveraging additional non-linear features on top of the simple ones used for the clean solution.

Finally, in what follows, we will present further evidence that CO is generally induced by features. Specifically, we will show that other feature manipulations besides the injection of features can also influence CO. Additionally, we will show that the primary strategies effective at preventing CO in natural datasets can also mitigate CO in our manipulated datasets. This suggests a commonality between these types of CO, further underscoring the widespread relevance of our mechanism

## 7 Analysis of CO prevention methods

Our proposed dataset intervention, defined in Section 3, has allowed us to better understand the chain of events that lead to CO. In this section, we will focus on methods to prevent CO and analyze them in our framework for further insights.

Table 1: Clean and robust accuracies of FGSM-AT and PGD-AT trained networks on CIFAR-10 and the low pass version described in Ortiz-Jimenez et al. (2020b) at different $\epsilon$.

| Method ($\epsilon$) | Original | | Low pass | |
|---|---|---|---|---|
| | Clean | Robust | Clean | Robust |
| FGSM ($8/255$) | 85.6 | 0.0 | 81.1 | 47.0 |
| PGD ($8/255$) | 80.9 | 50.6 | 80.3 | 49.7 |
| FGSM ($16/255$) | 76.3 | 0.0 | 78.6 | 0.0 |
| PGD ($16/255$) | 68.0 | 29.2 | 66.9 | 28.4 |

**High-frequency features and CO** Recently, Grabinski et al. (2022) suggested that CO could be caused by aliasing effects from downsampling layers and proposed using a low-pass filter before pooling. We take this theory a step further and show that just removing the high-frequency components from CIFAR-10 consistently prevents CO at $\epsilon = 8/255$ (where FGSM-AT fails). Furthermore, in Figure 6 (right) we observe that after CO on $\widetilde{\mathcal{D}}_\beta$, FGSM attacks ignore low-frequencies and become increasingly high-frequency.

Surprisingly, though, applying the same low-pass technique at $\epsilon = 16/255$ does not work and neither does Grabinski et al. (2022) (see Appendix J). We conjecture this is because the features robust at $\epsilon = 8/255$ in the low pass version of CIFAR-10 might not be robust at $\epsilon = 16/255$ and although high-frequency features might contribute to

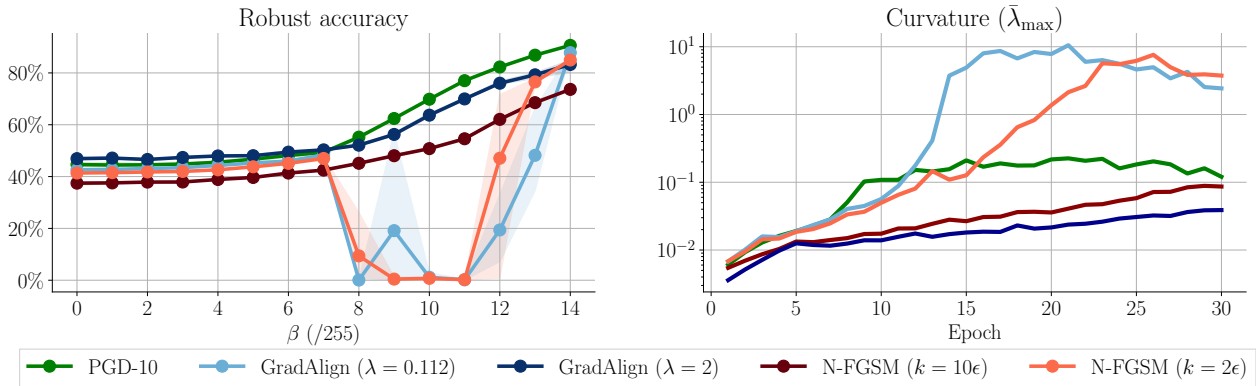

Figure 7: **Left:** Clean and robust accuracy of GradAlign, N-FGSM and PGD-AT. **Right:** Curvature evolution of all methods. Training performed on $\widetilde{\mathcal{D}}_\beta$ with $\epsilon = 6/255$ and $\beta = 8/255$ and averaged over 3 random seeds.

CO, the overall set of CO inducing features is more complex. Additionally, we observe that CO can be induced even when the injected features are randomly sampled (see Figure 14). Although removing low-frequency features does not work in all settings, we see this as a proof of concept that manipulating the data itself may prevent CO in some settings. Generalizing this idea to other datasets is a promising avenue for future work.

**Preventing CO in our injected dataset** In Figure 2 (left), we have shown that using PGD-AT prevents CO on our injected dataset, but we now also evaluate the effectiveness of GradAlign and N-FGSM. Our results indicate that these methods can also be successful in preventing CO, but need stronger regularization for certain values of $\beta$. As $\beta$ grows, the injected feature $\boldsymbol{v}(y)$ becomes more discriminative, creating a stronger incentive for the network to use it. This increases the chances of CO as the network increases its curvature to combine the injected features with others in order to become robust. Figure 7 shows that the curvature of N-FGSM and GradAlign decreases with stronger regularizations, similar to PGD-AT. This supports the idea that preventing the curvature from exploding can indeed prevent CO.

**Stronger CO on high-resolution datasets** As discussed in Section 3, we observe CO when injecting discriminative features in several datasets. Interestingly, for ImageNet-100, we have observed that even multi-step attacks like PGD are not able to find adversarial perturbations after CO (see Figure 8). However, when using AutoAttack (AA) (Croce & Hein, 2020), we find that models are not robust[4]. To the best of our knowledge, this had not been observed before with "vanilla" CO, where after CO, models would be vulnerable to very weak PGD attacks, like PGD with 10 steps (Wong et al., 2020). We argue that in this case, the curvature increase that breaks FGSM also hinders PGD optimization. This aligns with our findings with N-FGSM and GradAlign, where stronger regularization is needed to prevent CO with the injected datasets, and suggests that CO is *stronger* in $\widetilde{\mathcal{D}}_\beta$.

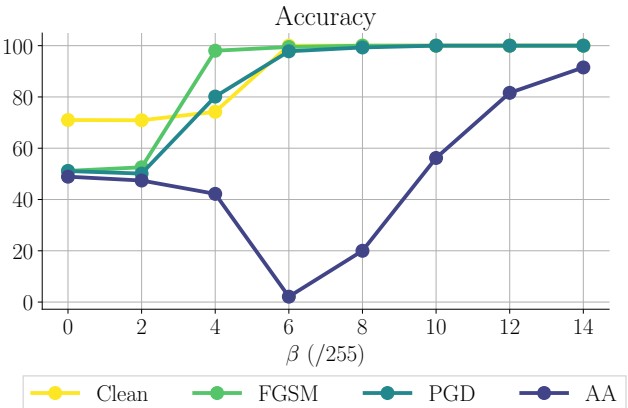

Figure 8: Clean, FGSM, PGD and AutoAttack accuracy on injected ImageNet-100 with varying $\beta$. FGSM-AT performed wtih $\epsilon = 4/255$.

---

[4]On other datasets, AA and PGD work equally.

# 8 Concluding remarks

In this work, we have presented a thorough empirical study to establish a link between the features of the data and the onset of CO in FGSM-AT. Using controlled data interventions, we have shown that CO is a learning shortcut used by the network to avoid learning complex robust features while achieving high accuracy using easy non-robust ones. This new perspective sheds new light on the mechanisms that trigger CO, as it shifts our focus towards studying the way the data structure influences the learning algorithm. We believe this opens the door to promising future work towards understanding the intricacies of these mechanisms, deriving methods for inspecting data and identifying feature interactions, and finding faster ways to make neural networks robust.

## Broader impact

The broader social implications of this research extend beyond its academic scope, touching upon the spheres of the entire field of adversarial machine learning. At its core, our work aims to enhance the robustness and reliability of AI systems, thereby addressing critical challenges in various sectors such as healthcare and autonomous driving. In these domains, mitigating the vulnerability to adversarial examples can lead to more accurate and reliable diagnoses and treatments, as well as safer navigational decisions during unforeseen events like storms.

Moreover, having a better understanding of CO has the potential to democratize access to robust machine learning models. By enabling faster adversarial training at a fraction of the cost, we could make advanced AI tools more accessible to smaller organizations and independent researchers. This democratization could spur innovation across various sectors, broadening the positive impacts of AI.

However, while these benefits are significant, it is crucial to acknowledge the potential drawbacks. An increase in the robustness of AI systems, while generally beneficial, can inadvertently strengthen systems employed for detrimental purposes or in exploitative circumstances, as illustrated by Albert et al. (2020; 2021). These paradoxical situations, such as those occurring under warfare conditions or within authoritarian regimes, underscore the fact that enhancing AI system robustness can sometimes lead to negative societal impacts.

Our work in this paper is purely academic, focusing on the technical understanding and prevention of CO. While we aim to improve AI systems' robustness and democratize access to them, it is the responsibility of policymakers, technologists, and society at large to ensure these advancements are employed ethically and for the greater good.

## Acknowledgements

We thank Maksym Andriushchenko, Apostolos Modas, Seyed-Mohsen Moosavi-Dezfooli and Ricardo Volpi for the fruitful discussions and feedback. This work is supported by the UKRI grant: Turing AI Fellowship EP/W002981/1 and EPSRC/MURI grant: EP/N019474/1. We would also like to thank the Royal Academy of Engineering and FiveAI. Guillermo Ortiz-Jimenez acknowledges travel support from ELISE (GA no 951847) in the context of the ELLIS PhD Program. Amartya Sanyal acknowledges partial support from the ETH AI Center.

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

# A    Analysis of the separability of the injected datasets

With the aim to illustrate how the interaction between $\mathcal{D}$ and $\mathcal{V}$ can influence the robustness of a classifier trained on $\widetilde{\mathcal{D}}_\beta$ we now provide a toy theoretical example in which we discuss this interaction. Specifically, without loss of generality, consider the binary classification setting on the dataset $(\boldsymbol{x}, y) \sim \mathcal{D}$ where $y \in \{-1, +1\}$ and $\|\boldsymbol{x}\|_2 = 1$, for ease. Let's now consider the injected dataset $\widetilde{\mathcal{D}}_\beta$ and further assume that $\boldsymbol{v}(+1) = \boldsymbol{u}$ and $\boldsymbol{v}(-1) = -\boldsymbol{u}$ with $\boldsymbol{u} \in \mathbb{R}^d$ and $\|\boldsymbol{u}\|_2 = 1$, such that $\widetilde{\boldsymbol{x}} = \boldsymbol{x} + \beta y \boldsymbol{u}$. Moreover, let $\gamma \in [0, 1]$ denote the *interaction coefficient* between $\mathcal{D}$ and $\mathcal{V}$, such that $-\gamma \leq \boldsymbol{x}^\top \boldsymbol{u} \leq \gamma$.

We are interested in characterizing the robustness of a classifier that only uses information in $\mathcal{V}$ when classifying $\widetilde{\mathcal{D}}_\beta$ depending on the strength of the interaction coefficient. In particular, as we are dealing with the binary setting, we will characterize the robustness of a linear classifier $h : \mathbb{R}^d \to \{-1, +1\}$ that discriminates the data based only on $\mathcal{V}$, *i.e.,* $h(\widetilde{\boldsymbol{x}}) = \text{sign}(\boldsymbol{u}^\top \widetilde{\boldsymbol{x}})$. In our setting, we have

$$\boldsymbol{u}^\top \widetilde{\boldsymbol{x}} = \boldsymbol{u}^\top \boldsymbol{x} + \beta \boldsymbol{u}^\top \boldsymbol{u} = \boldsymbol{u}^\top \boldsymbol{x} + \beta \quad \text{if } y = +1$$
$$\boldsymbol{u}^\top \widetilde{\boldsymbol{x}} = \boldsymbol{u}^\top \boldsymbol{x} - \beta \boldsymbol{u}^\top \boldsymbol{u} = \boldsymbol{u}^\top \boldsymbol{x} - \beta \quad \text{if } y = -1$$

**Proposition A.1** (Clean performance). *If $\beta > \gamma$, then $h$ achieves perfect classification accuracy on $\widetilde{\mathcal{D}}_\beta$.*

*Proof.* Observe that if $\gamma = 0$, i.e. the features from original dataset $\mathcal{D}$ do not interact with the injected features $\mathcal{V}$, the dataset is perfectly linearly separable. However, if the data $\boldsymbol{x}$ from $\mathcal{D}$ interacts with the injected signal $\boldsymbol{u}$, i.e. non zero projection, then the dataset is still perfectly separable but for a sufficiently larger $\beta$, such that $\boldsymbol{u}^\top \boldsymbol{x} + \beta > 0$ when $y = +1$ and $\boldsymbol{u}^\top \boldsymbol{x} + \beta < 0$ when $y = -1$. Because $-\gamma \leq \boldsymbol{x}^\top \boldsymbol{u} \leq \gamma$ this is achieved for $\beta > \gamma$. $\qquad\square$

**Proposition A.2** (Robustness). *If $\beta > \gamma$, the linear classifier $h$ is perfectly accurate and robust to adversarial perturbations in an $\ell_2$-ball of radius $\epsilon \leq \beta - \gamma$. Or, equivalently, for $h$ to be $\epsilon$-robust, the injected features must have a strength $\beta \geq \epsilon + \gamma$.*

*Proof.* Given $\widetilde{\boldsymbol{x}}$, we seek to find the minimum distance to the decision boundary of such a classifier. A minimum distance problem can be cast as solving the following optimization problem:

$$\epsilon^\star(\widetilde{\boldsymbol{x}}) = \min_{\boldsymbol{r} \in \mathbb{R}^d} \|\boldsymbol{r} - \widetilde{\boldsymbol{x}}\|_2^2 \ \text{ subject to } \boldsymbol{r}^\top \boldsymbol{u} = 0,$$

which can be solved in closed form

$$\epsilon^\star(\widetilde{\boldsymbol{x}}) = \frac{|\boldsymbol{u}^\top \widetilde{\boldsymbol{x}}|}{\|\boldsymbol{u}\|} = |\boldsymbol{u}^\top \boldsymbol{x} + y\beta|.$$

The robustness radius of the classifier $h$ will therefore be $\epsilon = \inf_{\widetilde{\boldsymbol{x}} \in \text{supp}(\widetilde{\mathcal{D}}_\beta)} \epsilon^\star(\widetilde{\boldsymbol{x}})$, which in our case can be bounded by

$$\epsilon = \inf_{(\widetilde{\boldsymbol{x}}, y) \in \text{supp}(\widetilde{\mathcal{D}}_\beta)} \epsilon^\star(\widetilde{\boldsymbol{x}}) \leq \min_{|\boldsymbol{u}^\top \boldsymbol{x}| \leq \gamma, y = \pm 1} |\boldsymbol{u}^\top \boldsymbol{x} + y\beta| = |\mp \gamma \pm \beta| = \beta - \gamma.$$

$\qquad\square$

Based on these propositions, we can clearly see that the interaction coefficient $\gamma$ reduces the robustness of the additive features $\mathcal{V}$. In this regard, if $\epsilon \geq \beta - \gamma$, robust classification at a radius $\epsilon$ can only be achieved by also leveraging information within $\mathcal{D}$.

# B   Robust classification can require non-linear features

We now provide a rigorous theoretical example of a learning problem that provably requires additional complex information for robust classification, even though it can achieve good clean performance using only simple features.

Given some $p \in \mathbb{N}$, let $\mathbb{R}^{p+1}$ be the input domain. A concept class, defined over $\mathbb{R}^{p+1}$ is a set of functions from $\mathbb{R}^{p+1}$ to $\{0, 1\}$. A hypothesis $h$ is $s$-non-linear if the polynomial with the smallest degree that can represent $h$ has a degree (largest order polynomial term) of $s$.

Using these concepts we now state the main result.

**Theorem 1.** *For any $p, k \in \mathbb{N}, \epsilon < 0.5$ such that $k < p$, there exits a family of distributions $\mathcal{D}_k$ over $\mathbb{R}^{p+1}$ and a concept class $\mathcal{H}$ defined over $\mathbb{R}^{p+1}$ such that*

1. *$\mathcal{H}$ is PAC learnable (with respect to the clean error) with a* linear *(degree 1)* *classifier. However, $\mathcal{H}$ is not robustly learnable with any linear classifier.*

2. *There exists an efficient learning algorithm, that given a dataset sampled i.i.d. from a distribution $\mathcal{D} \in \mathcal{D}_k$ robustly learns $\mathcal{H}$.*

*In particular, the algorithm returns a $k$-non-linear classifier and in addition, the returned classifier also exploits the linear features used by the linear non-robust classifier.*

*Proof.* We now define the construction of the distributions in $\mathcal{D}_k$. Every distribution $\mathcal{D}$ in the family of distribution $\mathcal{D}_k$ is uniquely defined by three parameters: a threshold parameter $\rho \in \{4t\epsilon : t \in \{0, \cdots, k\}\}$ (one can think of this as the non-robust, easy-to-learn feature), a $p$ dimensional bit vector $\boldsymbol{c} \in \{0, 1\}^p$ such that $\|\boldsymbol{c}\|_1 = k$ (this is the non-linear but robust feature) and $\epsilon$. Therefore, given $\rho$ and $\boldsymbol{c}$ (and $\epsilon$ which we discuss when necessary and ignore from the notation for simplicity), we can define the distribution $\mathcal{D}^{\boldsymbol{c}, \rho}$. We provide an illustration of this distribution for $p = 2$ in Figure 9.

**Sampling the robust non-linear feature**   To sample a point $(\boldsymbol{x}, y) \in \mathbb{R}^{p+1}$ from the distribution $\mathcal{D}^{\boldsymbol{c}, \rho}$, first, sample a random bit vector $\hat{\boldsymbol{x}} \in \mathbb{R}^p$ from the uniform distribution over the boolean hypercube $\{0, 1\}^p$. Let $\hat{y} = \sum_{i=1}^{p-1} \hat{\boldsymbol{x}}[i] \cdot \boldsymbol{c}[i] \pmod 2$ be the label of the parity function with respect to $\boldsymbol{c}$ evaluated on $\hat{\boldsymbol{x}}$. The marginal distribution over $\hat{y}$, if sampled this way, is equivalent to the Bernoulli distribution with parameter $\frac{1}{2}$. To see why, fix all bits in the input except one (chosen arbitrarily from the variables of the parity function), which is distributed uniformly over $\{0, 1\}$. It is easy to see that this forces the output of the parity function to be distributed uniformly over $\{0, 1\}$ as well. Repeating this process for all dichotomies of $p-1$ variables of the parity function proves the desired result.  Intuitively, $\hat{\boldsymbol{x}}$ constitutes the robust non-linear feature of this distribution.

**Sampling the non-robust linear feature**   To ensure that $\hat{\boldsymbol{x}}$ is not perfectly correlated with the true label, we sample the true label $y$ from a Bernoulli distribution with parameter $\frac{1}{2}$. Then we sample the non-robust feature $\boldsymbol{x}_1$ as follows

$$\boldsymbol{x}_1 \sim \begin{cases} \mathrm{Unif}\left(X_1^-\right) & y = 0 \wedge \hat{y} = 0 \\ \mathrm{Unif}\left(X_1^+\right) & y = 1 \wedge \hat{y} = 1 \\ \mathrm{Unif}\left(X_2^-\right) & y = 0 \wedge \hat{y} = 1 \\ \mathrm{Unif}\left(X_2^+\right) & y = 1 \wedge \hat{y} = 0 \end{cases}$$

where

$$X_1^+ = [\rho, \rho + \epsilon] \text{ and } \quad X_2^+ = [(\rho + 2\epsilon, \rho + 3\epsilon)], X_1^- = [\rho - \epsilon, \rho] \text{ and } \quad X_2^- = [(\rho - 3\epsilon, \rho - 2\epsilon)].$$

Finally, we return $(\boldsymbol{x}, y)$ where $\boldsymbol{x} = (\boldsymbol{x}_1; \hat{\boldsymbol{x}})$ is the concatenation of $\boldsymbol{x}_1$ and $\hat{\boldsymbol{x}}$.

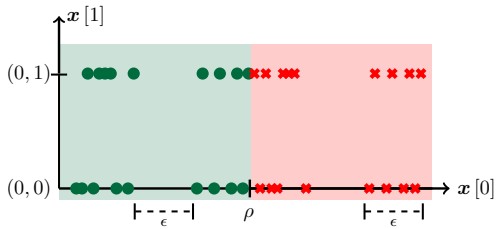
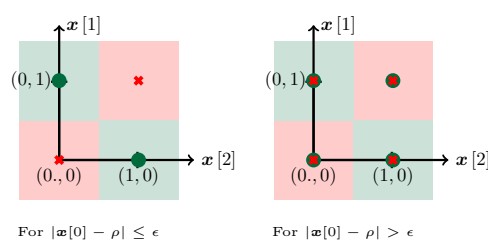

Figure 9: Illustration of one possible distribution $\mathcal{D}^{c,\rho}$ in three dimensions. The data is linearly separable in the direction $\boldsymbol{x}[0]$ but has a very smalll margin in that direction. Leveraging $\boldsymbol{x}[1]$ and $\boldsymbol{x}[2]$ additionally, we see how the data can indeed be separated robustly, albeit non-linearly.

**Linear non-robust solution**    First, we show that there is a linear, accurate, but non-robust solution to this problem. To obtain this solution, sample an $m$-sized dataset $S_m = \{(\boldsymbol{x}_1, y_1), \dots, (\boldsymbol{x}_m, y_m)\} \in \mathbb{R}^{p+1} \times \{-1, 1\}$ from the distribution $\mathcal{D}^{c,\rho}$. Ignore all, except the first coordinate, of the covariates of the dataset to create $S_m^0 = \{(\boldsymbol{x}_1[0], y_1) \dots (\boldsymbol{x}_m[0], y_m)\}$ where $S_i[j]$ indexes the $j^{th}$ coordinate of the $i^{th}$ element of the dataset. Then, sort $S_m^0$ on the basis of the covariates (*i.e.*, the first coordinate). Let $\hat{\rho}$ be the largest element whose label is 0.

Define $f_{\text{lin},\hat{\rho}}$ as the linear threshold function on the first coordinate i.e. $f_{\text{lin},\hat{\rho}}(\boldsymbol{x}) = \mathbb{I}\{\boldsymbol{x}[0] \geq \hat{\rho}\}$. By construction, $f_{\text{lin},\hat{\rho}}$ accurate classifies all points in $S_m$. The VC dimension of a linear threshold function in 1 dimension is 2. Then, using standard VC sample complexity upper bounds [5] for consistent learners, if $m \geq \kappa_0 \left(\frac{1}{\alpha} \log\left(\frac{1}{\beta}\right) + \frac{1}{\alpha} \log\left(\frac{1}{\alpha}\right)\right)$, where $\kappa_0$ is some universal constant, we have that with probability at least $1 - \beta$,

$$\text{Err}\left(f_{\text{lin},\hat{\rho}}; \mathcal{D}^{c,\rho}\right) \leq \alpha.$$

**Non-linear robust solution**    Next, we propose an algorithm to find a robust solution and show that this solution has a non-linearity of degree $k$. First, sample the $m$-sized dataset $S_m$ and use the method described above to find $\hat{\rho}$. Then, create a modified dataset $\widehat{S}$ by first removing all points $\boldsymbol{x}$ from $S_m$ such that $|\boldsymbol{x}[0] - \hat{\rho}| \geq \frac{\epsilon}{8}$ and then removing the first coordinate of the remaining points. Thus, each element in $\widehat{S}$ belongs to $\mathbb{R}^p \times \{0, 1\}$ dimensional dataset.

Note, that by construction, there is a consistent (*i.e.*, an accurate) parity classifier on $\widehat{S}$. Let the parity bit vector consistent with $\widehat{S}$ be $\hat{\boldsymbol{c}}$. This can be learned using Gaussian elimination. Consequently, construct the parity classifier $f_{\text{par},\widehat{c}} = \sum_{i=0}^{p-1} \widehat{\boldsymbol{x}}[i] \cdot \widehat{\boldsymbol{c}}[i] \pmod 2$.

Finally, the algorithm returns the classifier $g_{\hat{\rho},\hat{\boldsymbol{c}}}$, which acts as follows:

$$g_{\hat{\rho},\hat{\boldsymbol{c}}}(\boldsymbol{x}) = \begin{cases} 1 & \mathbb{I}\left\{\boldsymbol{x}[0] \geq \hat{\rho} + \epsilon + \frac{\epsilon}{8}\right\} \\ 0 & \mathbb{I}\left\{\boldsymbol{x}[0] \leq \hat{\rho} - \epsilon - \frac{\epsilon}{8}\right\} \\ f_{\text{par},\widehat{c}}(\widetilde{\boldsymbol{x}}) & \text{o.w.} \end{cases} \tag{5}$$

where $\widetilde{\boldsymbol{x}} = \text{round}(\boldsymbol{x}[1, \dots, p])$ is obtained by rounding off $\boldsymbol{x}$ starting from the second index till the last. For example, if $\boldsymbol{x} = [0.99, 0.4, 0.9, 0.4, 0.8]$, $\epsilon = 0.2$, and $\boldsymbol{c} = [0, 0, 1, 1]$ then $\widetilde{\boldsymbol{x}} = [0, 1, 0, 1]$ and $g_{0.5,\hat{\boldsymbol{c}}}[\widetilde{x}] = 1$. Finally, it is easy to verify that the classifier $g_{\hat{\rho},\hat{\boldsymbol{c}}}$ is accurate on all training points and as the number of total parity classifiers is less than $2^p$ (hence finite VC dimension), as long as $m \geq \kappa_1 \left(\frac{1}{\alpha} \log\left(\frac{1}{\beta}\right) + \frac{p}{\alpha} \log\left(\frac{1}{\alpha}\right)\right)$, where $\kappa_1$ is some universal constant, we have that with probability at least $1 - \beta$,

$$\text{Err}\left(g_{\hat{\rho},\hat{\boldsymbol{c}}}; \mathcal{D}^{c,\rho}\right) \leq \alpha.$$

**Robustness of $g_{\hat{\rho},\hat{\boldsymbol{c}}}$**    As $\boldsymbol{x}[0]$ is distributed uniformly in the intervals $[\rho - \epsilon, \rho] \cup [\rho, \rho + \epsilon]$, we have that $|\rho - \hat{\rho}| \leq 4\epsilon \cdot \text{Err}(f_{\text{lin},\hat{\rho}}; \mathcal{D}^{c,\rho}) \leq 4\epsilon\alpha$. Therefore, when $m$ is large enough $\left(m = \text{poly}\left(\frac{1}{\alpha}\right)\right)$ such that $\alpha \leq \frac{1}{32}$, we have that $|\hat{\rho} - \rho| \leq \frac{\epsilon}{8}$. Intuitively, this guarantees that $g_{\hat{\rho},\hat{\boldsymbol{c}}}$ uses the linear threshold function on $\boldsymbol{x}[0]$ for classification in the interval $[\rho, \rho + \epsilon] \cup [\rho, \rho - \epsilon]$ and $f_{\text{par},\widehat{c}}$ in the $[\rho + 2\epsilon, \rho + 3\epsilon] \cup [\rho - 2\epsilon, \rho - 3\epsilon]$. A crucial property

---

[5] https://www.cs.ox.ac.uk/people/varun/kanade/teaching/CLT-MT2018/lectures/lecture03.pdf

of $g_{\hat{\rho},\hat{c}}$ is that for all $x \in \mathrm{Supp}\left(\mathcal{D}^{c,\rho}\right)$, the classifier $g_{\hat{\rho},\hat{c}}$ does not alter its prediction in an $\ell_\infty$-ball of radius $\epsilon$. We show this by studying four separate cases. First, we prove robustness along all coordinates except the first.

1. When $|\boldsymbol{x}[0] - \hat{\rho}| \geq \epsilon + \frac{\epsilon}{8}$, as shown above, $g_{\hat{\rho},\hat{c}}$ is invariant to all $\boldsymbol{x}[i]$ for all $i > 0$ and is thus robust against all $\ell_\infty$ perturbations against those coordinates.

2. When $|\boldsymbol{x}[0] - \hat{\rho}| < \epsilon + \frac{\epsilon}{8}$, due to Equation (5), we have that $g_{\hat{\rho},\hat{c}} = f_{\mathrm{par},\hat{c}}(\widetilde{\boldsymbol{x}})$ where $\widetilde{\boldsymbol{x}} = \mathrm{round}\left(\boldsymbol{x}[1,\dots,p]\right)$ is obtained by rounding off all indices of $\boldsymbol{x}$ except the first. As the rounding operation on the boolean hypercube is robust to any $\ell_\infty$ perturbation of radius less than 0.5, we have that $g_{\hat{\rho},\hat{c}}$ is robust to all $\ell_\infty$ perturbations of radius less than 0.5 on the support of the distribution $\mathcal{D}^{c,\rho}$.

Next, we prove the robustness along the first coordinate. Let $0 < \delta < \epsilon$ represent an adversarial perturbation. Without loss of generality, assume that $\boldsymbol{x}[0] > \hat{\rho}$ as similar arguments apply for the other case.

1. Consider the case $\boldsymbol{x}[0] \leq \hat{\rho} + \epsilon + \frac{\epsilon}{8}$. Then, $|\boldsymbol{x}[0] - \delta - \hat{\rho}| \leq |\epsilon + \frac{\epsilon}{8} - \delta| \leq \epsilon + \frac{\epsilon}{8}$ and hence, by construction, $g_{\hat{\rho},\hat{c}}(\boldsymbol{x}) = g_{\hat{\rho},\hat{c}}([x[0] - \delta; [x][1,\dots,p]])$. On the other hand, for all $\delta$, we have that $g_{\hat{\rho},\hat{c}}([x[0] + \delta; [x][1,\dots,p]]) = 1$ if $g_{\hat{\rho},\hat{c}}(x) = 1$.

2. For the case $\boldsymbol{x}[0] \geq \hat{\rho} + \epsilon + \frac{\epsilon}{8}$, the distribution is supported only on the interval $[\rho + 2\epsilon, \rho + 3\epsilon]$. When a positive $\delta$ is added to the first coordinate, the classifier's prediction does not change and it remains 1. For all $\delta \leq \frac{\epsilon}{2}$, when the perturbation is subtracted from the first coordinate, its first coordinate is still larger than $\hat{\rho} + \epsilon + \frac{\epsilon}{8}$ and hence, the prediction is still 1.

This completes the proof of robustness of $g_{\hat{\rho},\hat{c}}$ along all dimensions to $\ell_\infty$ perturbations of radius less than $\epsilon$. Combining this with its error bound, we have that $\mathrm{Adv}_{\epsilon,\infty}\left(g_{\hat{\rho},\hat{c}}; \mathcal{D}^{c,\rho}\right) \leq \alpha$.

To show that the parity function is non-linear, we use a classical result from Aspnes et al. (1994). Theorem 2.2 in Aspnes et al. (1994) shows that approximating the parity function in $k$ bits using a polynomial of degree $\ell$ incurs at least $\sum_{i=0}^{k_\ell} \binom{k}{i}$ where $k_\ell = \lfloor \frac{k-\ell-1}{2} \rfloor$ mistakes. Therefore, the lowest degree polynomial that can do the approximation accurately is at least $k$.

This completes our proof of showing that the robust classifier is of non-linear degree $k$ while the accurate classifier is linear. Next, we prove that no linear classifier can be robust. We show this by contradiction.

**No linear classifier can be robust** Construct a set $\mathcal{Z}$ of $s$ (to be defined later) points in $\mathbb{R}^{p+1}$ by sampling the first coordinate from the interval $[\rho, \rho + \epsilon]$ and the remaining $p$ coordinates uniformly from the boolean hypercube. Then, augment the set by subtracting $\epsilon$ from the first coordinate while retaining the rest of the coordinates. Note that this set can be generated, along with its labels, by sampling enough points from the original distribution and discarding points that do not fall in this interval. Now construct adversarial examples of each point in the augmented set by either adding or subtracting $\epsilon$ from the negatively and the positively labelled examples respectively and augment the original set with these adversarial points. For a large enough $s$,[6] this augmented set of points can be decomposed into a multiset of points, where all points in any one set has the same value in the first coordinate but nearly half of their label is zero and the other half one.

Now, assume that there is a linear classifier that has a low error on the distribution $\mathcal{D}^{c,\rho}$. Therefore the classifier is also accurate on these sets of points as the classifier is robust, by assumption, and the union of these sets occupy a significant under the distribution $\mathcal{D}^{c,\rho}$. However, as the first coordinate of every point within a set is constant despite half the points having label one and the other half zero, the coefficient of the linear classifier can be set to zero without altering the behavior of the classifier. Then, effectively the linear classifier is representing a parity function on the rest of the $p$ coordinates. However, we have just seen that this is not possible as a linear threshold function cannot represent a parity function on $k$ bits where $k > 1$. This contradicts our initial assumption that there is a robust linear classifier for this problem.

This completes the proof. $\qquad\square$

---

[6]There is a slight technicality as we might not obtain points that are exact reflections of each other around $\rho$ but that can be overcome by discretising upto a certain precision

## C    Experimental details

In this section we provide the experimental details for all results presented in the paper. Adversarial training for all methods and datasets follows the fast training schedules with a cyclic learning rate introduced in Wong et al. (2020). We train for 30 epochs on CIFAR Krizhevsky & Hinton (2009) and 15 epochs for SVHN Netzer et al. (2011) following Andriushchenko & Flammarion (2020). When we perform PGD-AT we use 10 steps and a step size $\alpha = 2/255$; FGSM uses a step size of $\alpha = \epsilon$. Regularization parameters for GradAlign Andriushchenko & Flammarion (2020) and N-FGSM de Jorge et al. (2022) will vary and are stated when relevant in the paper. The architecture employed is a PreactResNet18 He et al. (2016). Robust accuracy is evaluated by attacking the trained models with PGD-50-10, i.e. PGD with 50 iterations and 10 restarts. In this case we also employ a step size $\alpha = 2/255$ as in Wong et al. (2020). All accuracies are averaged after training and evaluating with 3 random seeds.

The curvature computation is performed following the procedure described in Moosavi-Dezfooli et al. (2019). As they propose, we use finite differences to estimate the directional second derivative of the loss with respect to the input, *i.e.,*

$$\boldsymbol{w}^\top \nabla_{\boldsymbol{x}}^2 \mathcal{L}(f_{\boldsymbol{\theta}}(\boldsymbol{x}), y) \approx \frac{\nabla_{\boldsymbol{x}} \mathcal{L}(f_{\boldsymbol{\theta}}(\boldsymbol{x} + t\boldsymbol{w}), y) - \nabla_{\boldsymbol{x}} \mathcal{L}(f_{\boldsymbol{\theta}}(\boldsymbol{x} - t\boldsymbol{w}), y)}{2t},$$

with $t > 0$ and use the Lanczos algorithm to perform a partial eigendecomposition of the Hessian without the need to compute the full matrix. In particular, we pick $t = 0.1$.

All our experiments were performed using a cluster equipped with GPUs of various architectures. The estimated compute budget required to produce all results in this work is around $2,000$ GPU hours (in terms of NVIDIA V100 GPUs).

## D    Inducing catastrophic overfitting with other settings

In Section 3 we have shown that CO can be induced with data interventions for CIFAR-10 and $\ell_\infty$ perturbations. Here we present similar results when using other datasets (i.e. CIFAR-100 and SVHN) and other types of perturbations (i.e. $\ell_2$ attacks). Moreover, we also report results when the injected features $\boldsymbol{v}(y)$ follow random directions (as opposed to low-frequency DCT components). Overall, we find similar findings to those reported the main text.

### D.1    Other datasets

Similarly to Section 3 we modify the SVHN, CIFAR-100 and higher resolution Imagenet-100 and TinyImagenet datasets to inject highly discriminative features $\boldsymbol{v}(y)$. Since SVHN also has 10 classes, we use the exact same settings as in CIFAR-10 and we train and evaluate with $\epsilon = 4$ where training on the original data does not lead to CO (recall $\beta = 0$ corresponds to the unmodified dataset). On the other hand, for CIFAR-100 and Imagenet-100 we select $\boldsymbol{v}(y)$ to be the 100 DCT components with the lowest frequency and we present results with $\epsilon = 5$ and $\epsilon = 4$ respectively. Similarly, for TinyImagenet (which has 200 classes) we use the first 200 DCT components and present results with $\epsilon = 6$. On ImageNet-100 we evaluate robustness with AutoAttack Croce & Hein (2020), as PGD-50 seems to be broken when CO occurs in this dataset. On the other datasets, we evaluated with AutoAttack at the extrema of our plots and found that the estimated robustness agrees with the one of PGD-50. To save computational complexity, we only used PGD-50 to estimate the robustness on the other datasets.

Regarding the training settings, for CIFAR10/100 and SVHN datasets we use the same settings as Andriushchenko & Flammarion (2020). For ImageNet-100 we follow Kireev et al. (2022) and for TinyImageNet Li et al. (2020).

In all datasets we can observe similar trends as with CIFAR-10: For small values of $\beta$ the injected features are not very discriminative due to their interaction with the dataset images and the model largely ignores them. As we increase $\beta$, there is a range in which they become more discriminative but not yet robust and

we observe CO. Finally for large values of $\beta$ the injected features become robust and the models can achieve very good performance focusing only on those.

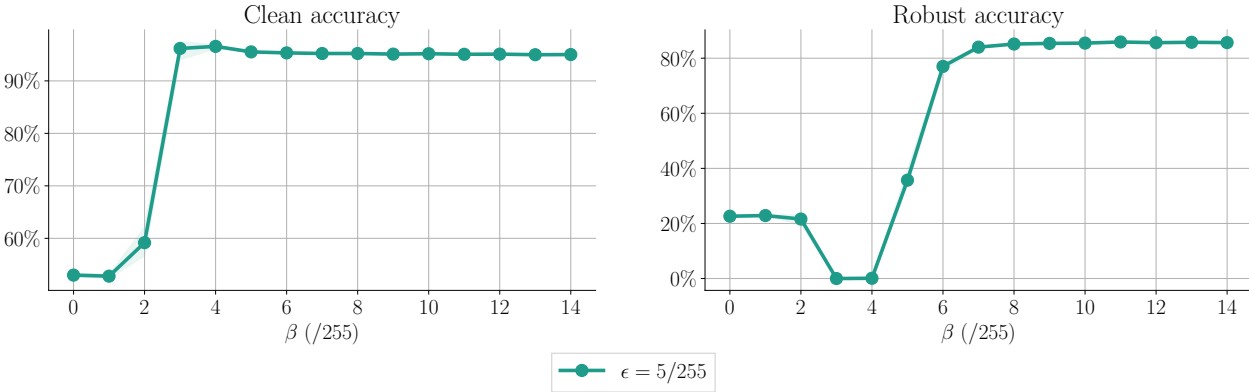

Figure 10: Clean and robust performance after FGSM-AT on injected datasets $\widetilde{\mathcal{D}}_\beta$ constructed from CIFAR-100. As FGSM-AT already suffers CO on CIFAR-100 at $\epsilon = {}^6/_{255}$ we use $\epsilon = {}^5/_{255}$ in this experiment where FGSM-AT does not suffer from CO as seen for $\beta = 0$. In this setting, we observe CO happening when $\beta$ is slightly smaller than $\epsilon$. Results are averaged over 3 seeds and shaded areas report minimum and maximum values.

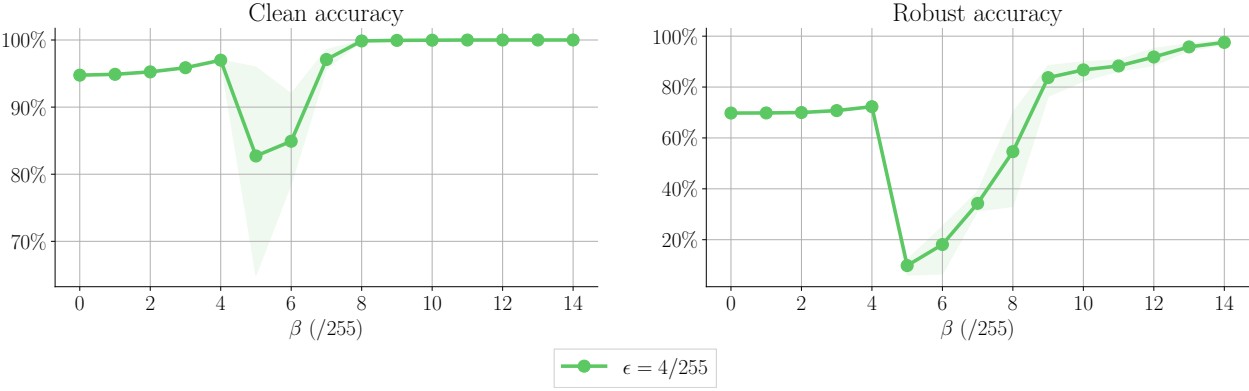

Figure 11: Clean and robust performance after FGSM-AT on injected datasets $\widetilde{\mathcal{D}}_\beta$ constructed from SVHN. As FGSM-AT already suffers CO on SVHN at $\epsilon = {}^6/_{255}$ we use $\epsilon = {}^4/_{255}$ in this experiment where FGSM-AT does not suffer from CO as seen for $\beta = 0$. In this setting, we observe CO happening when $\beta \approx \epsilon$. Results are averaged over 3 seeds and shaded areas report minimum and maximum values.

## D.2 Other norms

Catastrophic overfitting has been mainly studied for $\ell_\infty$ perturbations and thus we presented experiments with $\ell_\infty$ attacks following related work. However, in this section we also present results where we induce CO with $\ell_2$ perturbation which are also widely used in adversarial robustness. In Figure 13 we show the clean (left) and robust (right) accuracy after FGM-AT[7] on our injected dataset from CIFAR-10 ($\widetilde{\mathcal{D}}_\beta$). Similarly to our results with $\ell_\infty$ attacks, we also observe CO as the injected features become more discriminative (increased $\beta$). It is worth mentioning that the $\ell_2$ norm we use ($\epsilon = 1.5$) is noticeably larger than typically used in the literature, however, it would roughly match the magnitude of an $\ell_\infty$ perturbation with $\epsilon = 7/255$. Interestingly, we did not observe CO for this range of $\beta$ with $\epsilon = 1$.

---

[7]FGM is the $\ell_2$ version of FGSM where we do not take the sign of the gradient.

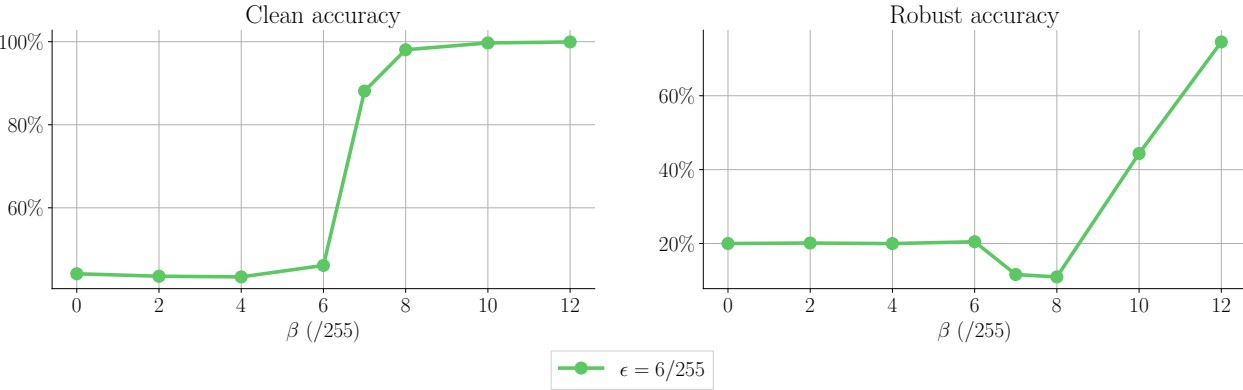

Figure 12: Clean and robust performance after FGSM-AT on injected datasets $\widetilde{\mathcal{D}}_\beta$ constructed from TinyImageNet. We use $\epsilon = {}^6/{}_{255}$ in this experiment where FGSM-AT does not suffer from CO as seen for $\beta = 0$. In this setting, we observe CO happening for $\beta \in [{}^7/{}_{255}, {}^8/{}_{255}]$.

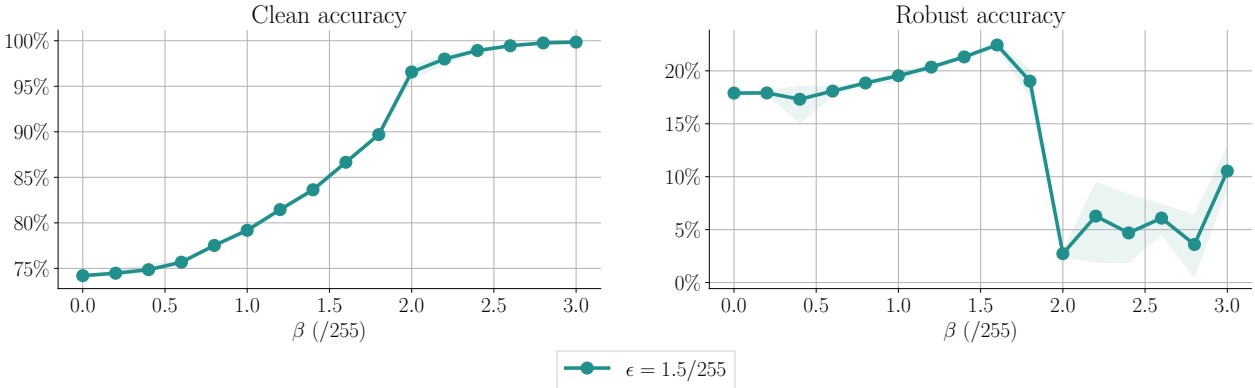

Figure 13: Clean and $\ell_2$ robust performance after FGSM-AT on injected datasets $\widetilde{\mathcal{D}}_\beta$ constructed from CIFAR-10. FGM-AT suffers CO on CIFAR-10 around $\epsilon = 2$, so we use $\epsilon = 1.5$ in this experiment where FGM-AT does not suffer from CO as seen for $\beta = 0$. In this setting, we observe CO happening when $\beta \approx \epsilon$. Results are averaged over 3 seeds and shaded areas report minimum and maximum values.

### D.3   Other injected features

We selected the injected features for our injected dataset from the low frequency components of the DCT to ensure an interaction with the features present on natural images Ahmed et al. (1974). However, this does not mean that other types of features could not induce CO. In order to understand how unique was our choice of features we also created another family of injected datasets but this time using a set of 10 randomly generated vectors as features. As in the main text, we take the sign of each random vector to ensure they take values in $\{-1, +1\}$ and assign one vector per class. In Figure 14 we observe that using random vectors as injected features we can also induce CO. Note that since our results are averaged over 3 random seeds, each seed uses a different set of random vectors.

### D.4   Other architectures

In all our previous experiments we trainde a PreActResNet18 (He et al., 2016) as it is the standard architecture used in the literature. However, our observations our also robust to the choice of architecture. As we can see in Figure 15, we can also induce CO when training a WideResNet28x10 Zagoruyko & Komodakis (2016) on an injected version of CIFAR-10.

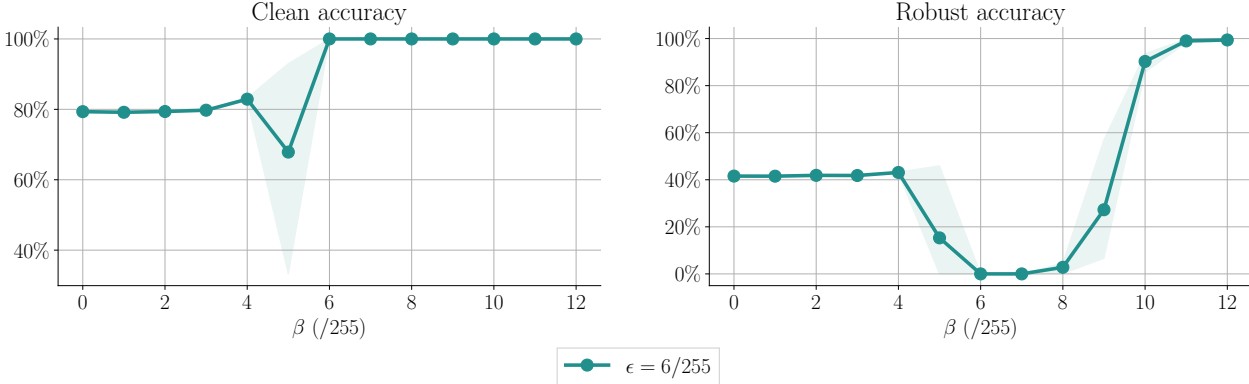

Figure 14: Clean and robust performance after FGSM-AT on injected datasets $\widetilde{\mathcal{D}}_\beta$ constructed from CIFAR-10 using random signals in $\mathcal{V}$. We perform this experiments at $\epsilon = {}^6\!/_{255}$ where we saw that injected the dataset with the DCT basis vectors did induce CO. In the random $\mathcal{V}$ setting, we observe the same behaviour, with CO happening when $\beta \approx \epsilon$. Results are averaged over 3 seeds and shaded areas report minimum and maximum values.

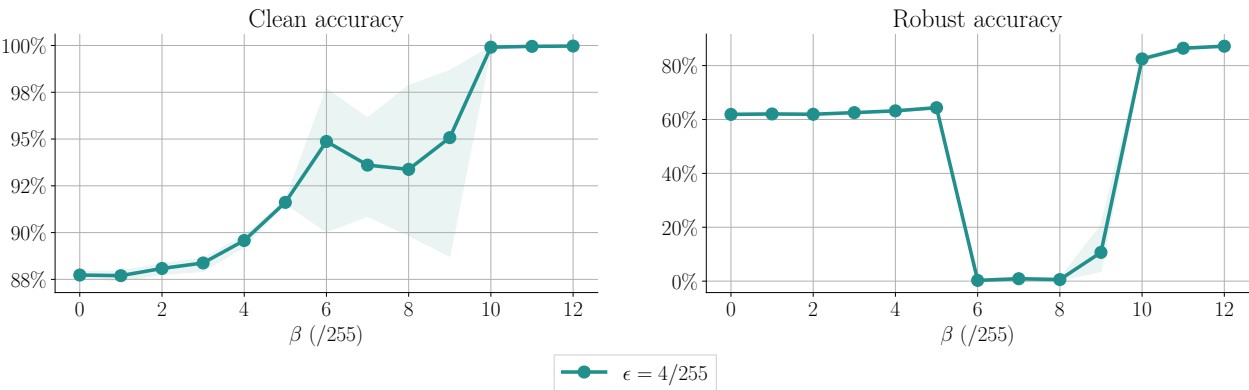

Figure 15: Clean and robust performance after FGSM-AT on injected datasets $\widetilde{\mathcal{D}}_\beta$ constructed from CIFAR-10 when training a WideResNet28x10. We perform this experiments at $\epsilon = {}^4\!/_{255}$. Results are averaged over 3 seeds and shaded areas report minimum and maximum values.

## E   Performance on $\mathcal{V}$

To further understand the behavior of the models described in Section 4, we conduct an additional experiment evaluating the performance of the networks on the set of features $\mathcal{V}$. This experiment corroborates the findings shown in Figure 2 and reveal more insights into how the networks utilize the features from $\mathcal{V}$ and $\mathcal{D}$.

In Figure 16, we see that networks trained using standard training demonstrate near-perfect accuracy on $\mathcal{V}$ from the start, indicating that they heavily rely on these features for classification and can correctly classify the data based solely on these. In contrast, networks trained with PGD-AT show a gradual improvement in accuracy on $\mathcal{V}$, as they learn to effectively combine the information from $\mathcal{V}$ with that of $\mathcal{D}$ to achieve robustness. As anticipated, networks trained with FGSM-AT exhibit a similar behavior to those trained with PGD-AT initially. However, following the occurrence of catastrophic overfitting (CO), they experience a sudden increase in accuracy on $\mathcal{V}$.

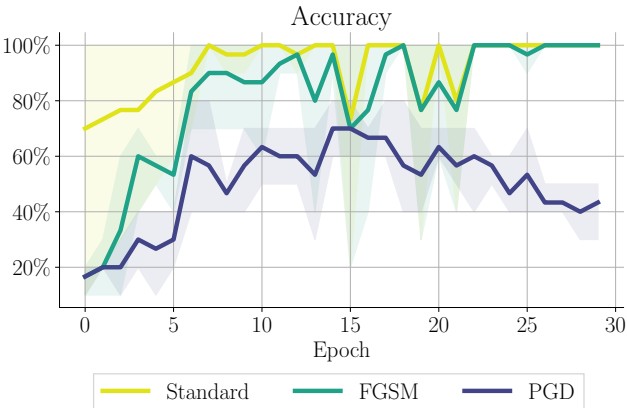

Figure 16: Accuracy on $\mathcal{V}$ of networks trained on $\widetilde{\mathcal{D}}_\beta$ using standard training, FGSM-AT or PGD-AT FGSM-AT. We use $\epsilon = {}^4\!/_{255}$ and $\beta = {}^6\!/_{255}$. Results averaged over 3 random seeds. Shades show min-max values.

## F  Learned features at different $\beta$

In Section 3 we discussed how, based on the strength of the injected features $\beta$, our injected datasets seem to have 3 distinct regimes: (i) When $\beta$ is small we argued that the network would not use the injected features as these would not be very helpful. (ii) When $\beta$ would have a very large value then the network would only look at these features since they would be easy-to-learn and provide enough margin to classify robustly. (iii) Finally, there was a middle range of $\beta$ usually when $\beta \sim \epsilon$ where the injected features would be strongly discriminative but not enough to provide robustness on their own. This regime is where we observe CO.

In this section we present an extension of Figure 2 where we take FGSM trained models on the injected datasets ($\widetilde{\mathcal{D}}_\beta$) and evaluate them on three test sets: (i) The injected test set ($\widetilde{\mathcal{D}}_\beta$) with the same features as the training set. (ii) The original dataset ($\mathcal{D}$) where the images are unmodified. (iii) The shuffled dataset ($\widetilde{\mathcal{D}}_{\pi(\beta)}$) where the injected features are permuted. That is, the set of injected features is the same but the class assignments are shuffled. Therefore, the injected features will provide conflicting information with the features present on the original image.

In Figure 17 we show the performance on the aforementioned datasets for three different values of $\beta$. For $\beta = {}^2\!/_{255}$ we are in regime (i) : we observe that the tree datasets have the same performance, i.e. the information of the injected features does not seem to alter the result. Therefore, we can conclude the network is mainly using the features from the original dataset $\mathcal{D}$. When $\beta = {}^{20}\!/_{255}$ we are in regime (ii) : the clean and robust performance of the network is almost perfect on the injected test set $\widetilde{\mathcal{D}}_\beta$ while it is close to 0% (note this is worse than random classifier) for the shuffled dataset. So when the injected and original features present conflicting information the network aligns with the injected features. Moreover, the performance on the original dataset is also very low. Therefore, the network is mainly using the injected features. Lastly, $\beta = {}^8\!/_{255}$ corresponds to regime (iii) : as discussed in Section 4, in this regime the network initially learns to combine information from both the original and injected features. However, after CO, the network seems to focus only on the injected features and discards the information from the original features.

## G  Analysis of curvature in different settings

In Figure 3 (left) we track the curvature of the loss surface while training on different injected datasets with either PGD-AT or FGSM-AT. We observe that (i) Curvature rapidly increases both for PGD-AT and FGSM-AT during the initial epochs of training. (ii) In those runs that presented CO, the curvature explodes around the $8^{th}$ epoch along with the training accuracy. (iii) When training with the dataset with orthogonally injected features ($\widetilde{\mathcal{D}}_\beta^\perp$) the curvature does not increase. This is aligned with our proposed mechanisms to induce CO whereby the network increases the curvature in order to combine different features to learn better

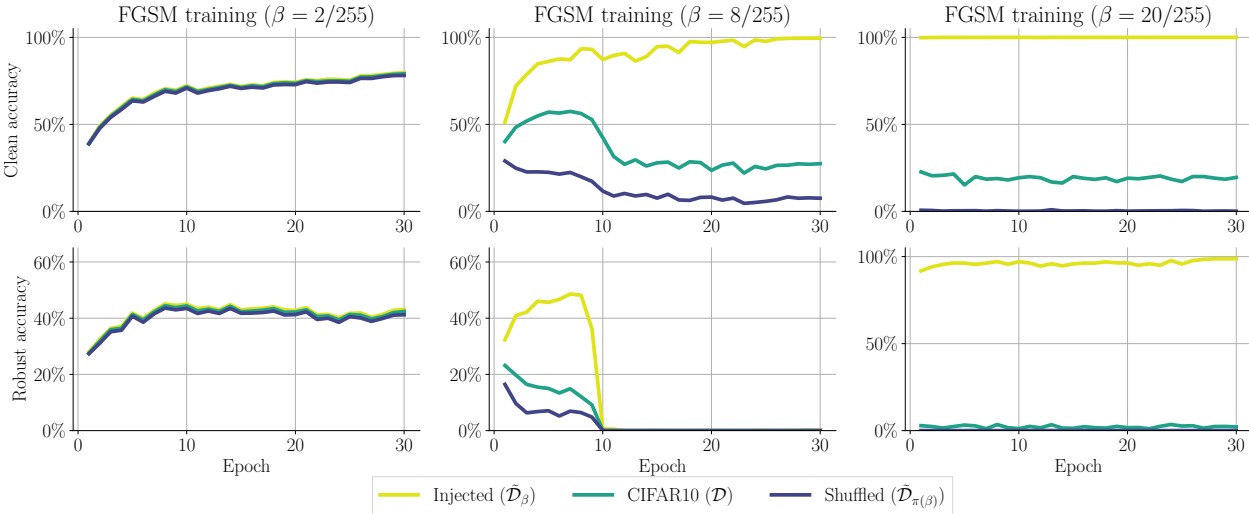

Figure 17: Clean (**top**) and robust (**bottom**) accuracy of FGSM-AT on $\widetilde{\mathcal{D}}_\beta$ at different $\beta$ values on 3 different test sets: (i) the original CIFAR-10 ($\mathcal{D}$), (ii) the dataset with injected features $\widetilde{\mathcal{D}}_\beta$ and (iii) the dataset with shuffled injected features $\widetilde{\mathcal{D}}_{\pi(\beta)}$. All training runs use $\epsilon = {}^6/_{255}$. **Left**: $\beta = {}^2/_{255}$ **Center**: $\beta = {}^8/_{255}$ **Right**: $\beta = {}^{20}/_{255}$.

representations. In this section we extend this analysis to different values of feature strength $\beta$ on the injected dataset ($\widetilde{\mathcal{D}}_\beta$). For details on how we estimate the curvature refer to Appendix C.

Figure 18 presents the curvature for different values of feature strength $\beta$ on the injected dataset ($\widetilde{\mathcal{D}}_\beta$). We show three different values of $\beta$ representative of the three regimes discussed in Appendix F. Recall that when $\beta$ is small ($\beta = {}^2/_{255}$) we observe that the model seems to focus only on CIFAR-10 features. Thus, we observe a curvature increase aligned with (CIFAR-10) feature combination. However, since for the chosen robustness radii $\epsilon = {}^6/_{255}$ there is no CO, we observe that the curvature increase remains stable. When $\beta$ is quite large ($\beta = {}^{20}/_{255}$) then the model largely ignores CIFAR-10 information and focuses on the easy-to-learn injected features. Since these features are already robust, there is no need to combine them and the curvature does not need to increase. In the middle range when CO happens ($\beta = {}^8/_{255}$) we observe again the initial curvature increase and then curvature explosion.

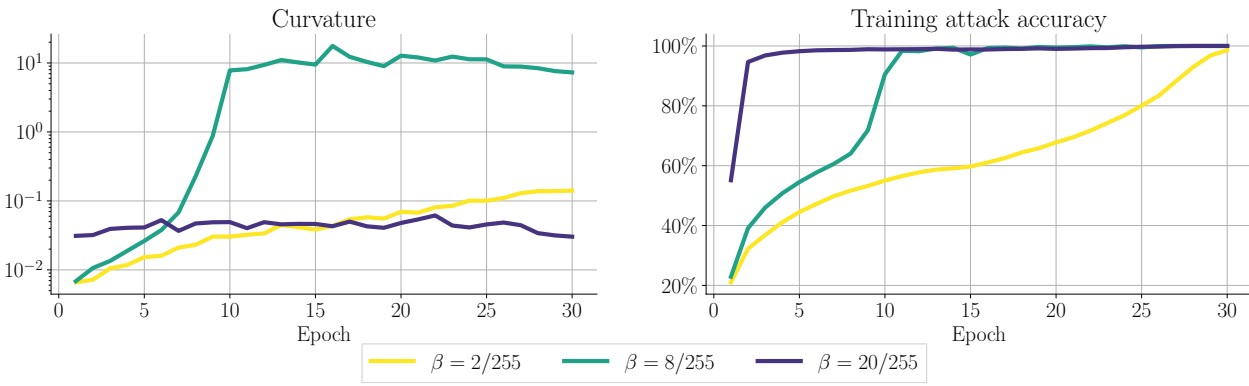

Figure 18: Evolution of curvature and training attack accuracy of FGSM-AT and PGD-AT trained on $\widetilde{\mathcal{D}}_\beta$ at different $\beta$ and for $\epsilon = {}^6/_{255}$. Only when CO happens (for $\beta = {}^8/_{255}$) the curvature explodes. For the other two interventions the curvature does not increase so much. We argue this is because the network does not need to disentangle $\mathcal{D}$ from $\mathcal{V}$, as it ignores either one of them.

# H    Adversarial perturbations before and after CO

**Qualitative analysis**   In order to further understand the change in behaviour after CO we presented visualizations of the FGSM perturbations before and after CO in Figure 4. We observed that while prior to CO, the injected feature components $\boldsymbol{v}(y)$ were clearly identifiable, after CO the perturbations do not seem to point in those directions although the network is strongly relying on them to classify. In Figure 19 and Figure 20 we show further visualizations of the perturbations obtained both with FGSM or PGD attacks on networks trained with either PGD-AT or FGSM-AT respectively.

We observe that when training with PGD-AT, i.e. the training does not suffer from CO, both PGD and FGSM attacks produce qualitatively similar results. In particular, all attacks seem to target the injected features with some noise due to the interaction with the features from CIFAR-10. For FGSM-AT, we observe that at the initial epochs (prior to CO) the pertubations are similar to those of PGD-AT, however, after CO perturbations change dramatically both for FGSM and PGD attacks. This aligns with the fact that the loss landscape of the network has dramatically changed, becoming strongly non-linear. This change yields single-step FGSM ineffective, however, the network remains vulnerable and multi-step attacks such as PGD are still able to find adversarial examples, which in this case do not point in the direction of discriminative features Jetley et al. (2018); Ilyas et al. (2019).

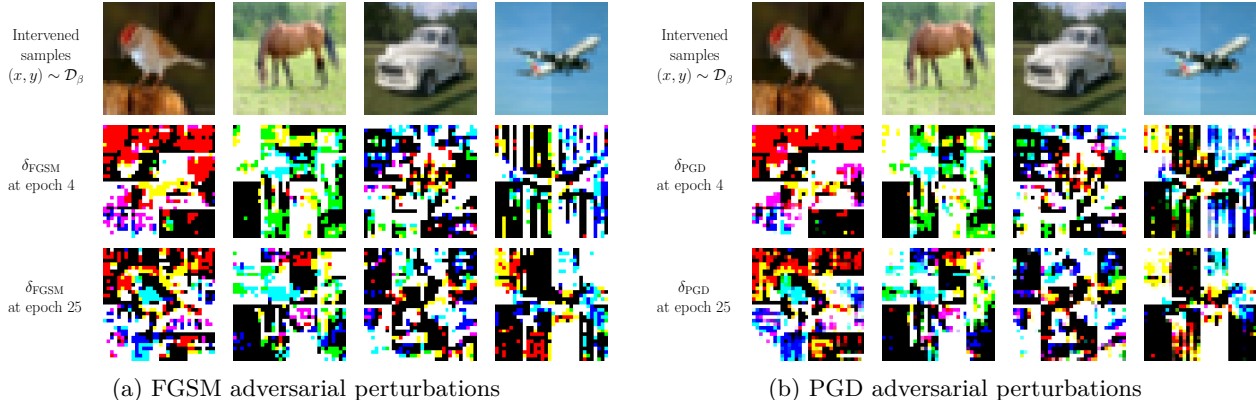

(a) FGSM adversarial perturbations                    (b) PGD adversarial perturbations

Figure 19: Different samples of the injected dataset $\widetilde{\mathcal{D}}_\beta$, and adversarial perturbations at epoch 4 and 22 of PGD-AT on $\widetilde{\mathcal{D}}_\beta$ at $\epsilon = 6/255$ and $\beta = 8/255$ (where FGSM-AT suffers CO). The adversarial perturbations remain qualitatively similar throughout training and align significantly with $\mathcal{V}$.

**Quantitative analysis**   Finally, to quantify the radical change of direction of the adversarial perturbations after CO, we compute the evolution of the average alignment (*i.e.,* cosine angle) between the FGSM perturbations $\boldsymbol{\delta}$ and the injected features, such that if point $\boldsymbol{x}$ is associated with class $y$ we compute $\frac{\langle \boldsymbol{\delta}, \boldsymbol{v}(y) \rangle}{\|\boldsymbol{v}(y)\|_2 \|\boldsymbol{\delta}\|_2}$. Figure 6 (left) shows the results of this evaluation, where we can see that before CO there is a non-trivial alignment between the FGSM perturbations and their corresponding injected features, that after CO quickly converges to the same alignment as the one between two random vectors.

To complement this view, we also perform an analysis of the frequency spectrum of the FGSM perturbations. In Figure 6 (right), we plot the average magnitude of the DCT transform of the FGSM perturbations computed on the test set of an intervened version of CIFAR-10 during training. As we can see, prior to CO, most of the energy of the perturbations is concentrated around the low frequencies (remember that the injected features are low frequency), but after CO happens, around epoch 8, the energy of the perturbations quickly gets concentrated towards higher frequencies. These two plots corroborate, quantitatively, our previous observations, that before CO, FGSM perturbations are pointing towards meaningful predictive features, while after CO, although we know the network still uses the injected features (see Figure 2) the FGSM perturbations suddenly point in a different direction.

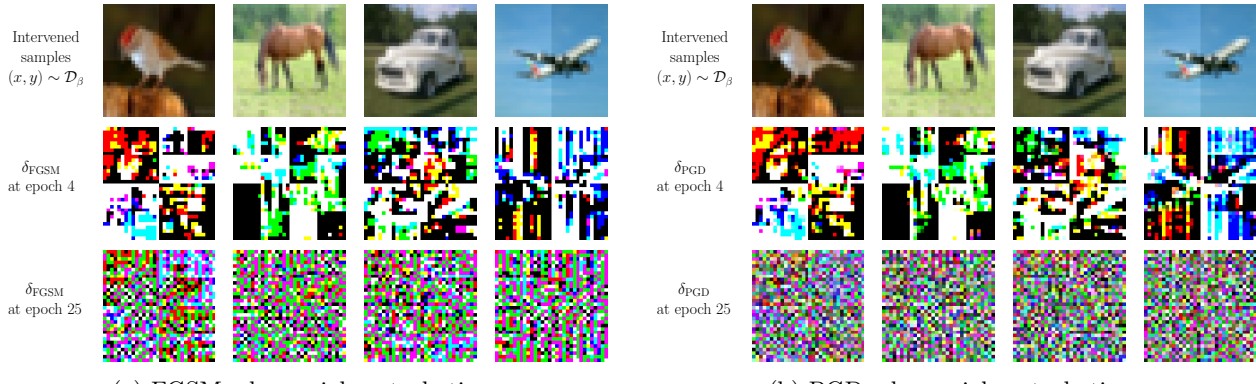

(a) FGSM adversarial perturbations

(b) PGD adversarial perturbations

Figure 20: Different samples of the injected dataset $\widetilde{\mathcal{D}}_\beta$, and adversarial perturbations at epoch 4 (before CO) and 22 (after CO) of FGSM-AT on $\widetilde{\mathcal{D}}_\beta$ at $\epsilon = {}^6\!/_{255}$ and $\beta = {}^8\!/_{255}$ (where FGSM-AT suffers CO). The adversarial perturbations change completely before and after CO. Prior to CO, they align significantly with $\mathcal{V}$, but after CO they point to meaningless directions.

# I  Further results with N-FGSM, GradAlign and PGD

In Section 7 we studied different SOTA methods that have been shown to prevent CO. Interestingly, we observed that in order to avoid CO on the injected dataset a stronger level of regularization is needed. Thus, indicating that the intervention is strongly favouring the mechanisms that lead to CO. For completeness, in Figure 21 we also present results of the clean accuracy (again with the robust accuracy). As expected, for those runs in which we observe CO, clean accuracy quickly saturates. Note that for stronger levels of regularization the clean accuracy is lower. An ablation of the regularization strength might help improve results further, however the purpose of this analysis is not to improve the performance on the injected dataset but rather to show it is indeed possible to prevent CO with the same methods that work for unmodified datasets.

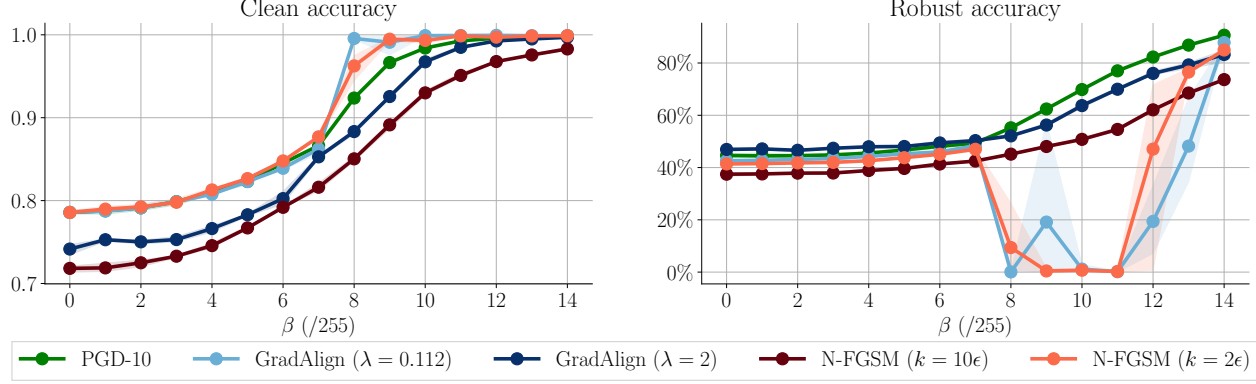

Figure 21: Clean (**left**) and robust (**right**) accuracy after AT with PGD-10, GradAlign and N-FGSM on $\widetilde{\mathcal{D}}_\beta$ at $\epsilon = {}^6\!/_{255}$. Results averaged over three random seeds and shaded areas report minimum and maximum values.

# J  Further details of low-pass experiment

We expand here over the results in Section 7 and provide further details on the experimental settings of Table 1. Specifically, we replicate the same experiment, *i.e.,* training a low-pass version of CIFAR-10 using FGSM-AT at different filtering bandwidths. As indicated in Section 7 we use the low-pass filter introduced

in Ortiz-Jimenez et al. (2020b) which only retains the frequency components in the upper left quadrant of the DCT transform of an image. That is, a low-pass filter of bandwidth $W$ would retain the $W \times W$ upper quadrant of DCT coefficients of all images, setting the rest of the coefficients to 0.

Figure 22 shows the robust accuracy obtained by FGSM-AT on CIFAR-10 versions that have been pre-filtered using such a low-pass filter. Interestingly, while training on the unfiltered images does induce CO on FGSM-AT, just removing a few high-frequency components is enough to prevent CO $\epsilon = 8255$. However, as described before, it seems that at $\epsilon = {}^{16}\!/_{255}$ no frequency transformation can avoid CO. Clearly, this transformation cannot be used as technique to prevent CO, but it highlights once again that the structure of the data plays a significant role in inducing CO.

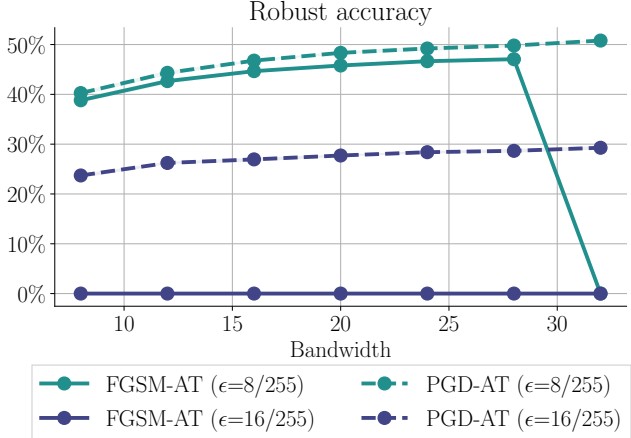

Figure 22: Robust accuracy of FGSM-AT and PGD-AT on different low-passed versions of CIFAR-10 using the DCT-based low pass filter introduced in Ortiz-Jimenez et al. (2020b). Bandwidth = 32 corresponds to the original CIFAR-10, while smaller bandwidths remove more and more high-frequency components. At $\epsilon = 8/255$ just removing a few high-frequency components is enough to prevent CO, while at $\epsilon = 16/255$ no frequency transformation avoids CO.

**Relation with anti-aliasing pooling layers**  As mentioned in Section 7, our proposed low-passing technique is very similar in spirit to works which propose using anti-aliasing low-pass filters at all pooling layers (Grabinski et al., 2022; Zhang, 2019). Indeed, as shown by Ortiz-Jimenez et al. (2020b), CIFAR10 contains a significant amount of non-robust features on the high-frequency end of the spectrum due to aliasing produced in their down-sampling process. In this regard, it is no surprise that methods like the one proposed in Grabinski et al. (2022) can prevent CO at $\epsilon = {}^{8}\!/_{255}$ using the same training protocol as in our work (robust accuracy is 45.9%). Interestingly, though, repeating the experiments in Grabinski et al. (2022) work using $\epsilon = {}^{16}\!/_{255}$ does lead to CO (robust accuracy is 0.0%) in Section 7. This result was not reported in the original paper, but we see it as a corroboration of our observations. Indeed, features play a role in CO, but the problematic features do not always come from excessively high-frequencies or aliasing. However, we still consider that preventing aliasing in the downsampling layers is a promising avenue for future work in adversarial robustness.

