# OpenReview forum: "Catastrophic overfitting can be induced with discriminative non-robust features"
_TMLR — Accepted by TMLR_

### Review · Reviewer_VSzM · 2023-05-17

**Summary Of Contributions:**

This paper focuses on catastrophic overfitting (CO) in single-step AT methods. The author proposes a new approach that introduces class-wise linearly separable noise into the natural dataset to investigate the relationship between CO, the magnitude of the noise added to the dataset, and the size of the attack budget used during training. Through a series of experiments, the author discovers that injecting noise into the dataset can induce CO. The authors have shown that CO is a kind of shortcut used by the network to avoid learning complex robust features while achieving high accuracy using easy non-robust ones. Furthermore, the authors investigate the mechanism of CO from the perspectives of curvature and feature interactions.


**Audience:**

Yes

**Broader Impact Concerns:**

There are no obvious concerns about the ethical implications of this work.

**Claims And Evidence:**

Yes

**Requested Changes:**

I wonder whether an experiment with ViT can be performed with the CO.

**Strengths And Weaknesses:**

Strengths:
1. The idea and method proposed by the author are innovative. Through experimental validation, it is demonstrated that CO can be induced, thereby inferring that the occurrence of CO in the original dataset is due to the same reasons. That is, the network exhibits a preference for certain features in the dataset, resulting in shortcuts.
2. The experiments are well-designed and comprehensive.
3. The writing is clear and easy to follow.

Weakness
1. As stated in Section 5.1, “before CO, the adversarial perturbations point in the direction of V.” But should it be in the opposite direction of V？ Since adversarial attack aims to obfuscate useful features and V is exactly the useful feature here. I believe some clarification is needed here.
2. There is a writing error in equation (3), the comma should be a multiplication symbol.

Post rebuttal:
Thank the authors for the clarification. I personally think the proposed finding is new and the authors have adequately verified the finding. I thus recommend acceptance of this paper. Also, I would like to suggest the authors add the keywords "adversarial" or "adversarial robustness" in the title to clarify the scope of the paper (as the phenomenon may not exist for other settings, like self-supervised training).

---

> ### Author Response · Authors · 2023-05-31
> **Reply to Reviewer VSzM**
>
> We would like to thank the reviewer for their time and kind words about our work. We appreciate that they found our work “*innovative*”, “*well-designed*”, “*comprehensive*”, and “*easy to follow*”.
>
> Below we address the main concerns of the reviewer:
>
> > Directionality of adversarial perturbations before CO
>
> The text in section 5.1 was indeed ambiguous, as in general we cannot claim that the adversarial perturbations point in the direction of V or opposite to it. In fact, we can at most expect that the adversarial perturbations have a non-trivial projection in $\operatorname{span}(\mathcal{V})$ (see Fig. 6) as adding vectors from $\operatorname{span}(\mathcal{V})$ to most natural images would in general modify the predicted class. For example, given a “dog” image, one could add the vector corresponding to the “deer” class or the “frog” class, and even a combination of both and probably fool a classifier that mostly relies on $\mathcal{V}$. Similarly, one could also subtract the vector associated with the class “dog” and also fool the classifier.
>
> We have modified the text in Section 5.1 to clarify this point.
>
> > Typo in Eq. (3)
>
> We have fixed the typo in the new version of the manuscript.
>
> > I wonder whether an experiment with ViT can be performed with the CO.
>
> We understand that recently ViT’s have become a widely popular architecture and acknowledge the value of the reviewer’s suggestion. However, we would like to politely ask the reviewer to consider if these experiments are essential for the publication of this manuscript given they are quite expensive. To motivate this further:
>
> Our primary aim in this work was to elucidate the specific conditions that induce CO by identifying a causal connection between data features and the emergence of CO. With this objective, our study has been focused on the most extensively examined fast-AT settings, which, to the best of our knowledge, have predominantly been proposed for classical CNNs.
> We have presented results using the two most popular architectures within this scope (PreActResNet18 and WideResNet28x10) and reproduced our experiments across five distinct datasets (CIFAR10, CIFAR100, SVHN, TinyImageNet and ImageNet100). These results have been corroborated with multiple random seeds and various types of injected features, culminating in a substantial total of more than 2,000 GPU hours. An integral aspect of managing this extensive experimental design was the availability of well-tuned hyperparameters for the fast-AT baselines from prior literature, but integrating ViTs into our experimental design would necessitate a substantial amount of time and resources to fine-tune them for acceptable performance in low-data settings and within adversarial training environments. This is still an ongoing developing area of research, as noted in recent literature [1-3], and would require a significant effort.
> That being said, should the reviewer think otherwise we would be open to discuss this further.
>
> [1] Chang Liu, Yinpeng Dong, Wenzhao Xiang, Xiao Yang, Hang Su, Jun Zhu, Yuefeng Chen, Yuan He, Hui Xue, Shibao Zheng. A Comprehensive Study on Robustness of Image Classification Models: Benchmarking and Rethinking. arXiv 2023
>
> [2] Naman D Singh, Francesco Croce, Matthias Hein. Revisiting Adversarial Training for ImageNet: Architectures, Training and Generalization across Threat Models. arXiv 2023
>
> [3] Edoardo Debenedetti, Vikash Sehwag, Prateek Mittal. A Light Recipe to Train Robust Vision Transformers. IEEE SaTML 2023

---

### Review · Reviewer_z6NM · 2023-05-24

**Summary Of Contributions:**

This paper experimentally explores the reason for catastrophic overfitting (CO), a problematic behavior of adversarial training utilizing single-step attacks.

The main experiment starts from injecting synthesized perturbations $\mathbf{v}(y)\in\mathcal{V}$ (of size $\beta$) that depend on the label $y$ to the training set (following the distribution \mathcal{D}). The authors first observe that for mildly strong perturbations of size $\beta\approx\epsilon$, they can induce CO for adversarial training using FGSM of attack size $\epsilon$.

Then, to understand why FGSM-based adversarial training (FGSM-AT) suffers CO for such $\beta\approx\epsilon$, the authors compare the clean/robust accuracies of standard training, FGSM-AT, and PGD-AT on three different datasets: the clean dataset, the dataset after adding label-dependent perturbations $\mathbf{v}(y)$, and the dataset after adding perturbations $\mathbf{v}(\pi(y))$ that depends on perturbed label $\pi(y)$. From this comparison, the authors observe that the FGSM-AT’s accuracies behave similarly to those of PGD-AT before CO, but they behave similarly to those of standard training after CO. Because standard training resorts to the label-dependent features $\mathcal{V}$ whereas PGD-AT utilizes both $\mathcal{D}$ and $\mathcal{V}$ to achieve high robust accuracies, they conclude that CO changes the learned feature of FGSM-AT from "both $\mathcal{D}$ and $\mathcal{V}$" to "$\mathcal{V}$ only".

Next, following the previous observations that the local curvature of the loss landscape explodes after CO, the authors tracked the evolutions of the loss curvatures for both FGSM-AT and PGD-AT. From the experiment, they found out that the FGSM-AT experience the explosion of loss curvature, whereas PGD-AT successfully suppresses the curvature increase. After the curvature increase, the local loss landscape becomes highly nonlinear. However, because FGSM assumes a linear loss landscape, FGSM starts generating meaningless perturbations for adversarial training, so FGSM-AT starts suffering CO. With an additional experiment with a dataset by projecting $\mathcal{D}$ onto the orthogonal complement of $\mathcal{V}$ (to remove features in $\mathcal{D}$ that interacts with $\mathcal{V}$), the authors also demonstrated that the curvature increase is due to the interactions between the features from $\mathcal{D}$ and those from $\mathcal{V}$.

Finally, the authors summarize their findings about the reason for CO, then present some additional analysis of existing CO prevention methods.


**Audience:**

Yes

**Broader Impact Concerns:**

I don’t see a particular broader impact concern regarding this paper.

**Claims And Evidence:**

Yes

**Requested Changes:**

1. As pointed out in the weakness, the experiment design may reflect something other than what really happens in real-world CO.
    1. I’m unsure whether it is possible to provide a connection between the settings of the experiments and the real-world scenario.
    2. Please justify that the findings can generalize to the realistic CO cases.
2. Figure 2 successfully visualizes the effect of CO on FGSM perturbations.
    1. Similarly, some people would be interested in looking at the effect of CO on the classification. To explain, after the model learns with the meaningless FGSM perturbations, does the model just lose robustness, or it starts focusing on the meaningless FGSM perturbation? (Is the FGSM perturbation just meaningless, or it has some negative impact on training?)
    2. To answer this question, I suggest the authors use pixel attribution methods such as Grad-CAM to highlight the feature that the model used for classifications.
3. The finding in the last paragraph of Section 7 may require more attention and raises other research questions.
    1. Do the authors have some experimental support that the curvature increase (by AutoAttack) actually hindered PGD optimization, or is there any other reason for this?
    2. If AutoAttack (AA) introduced such a bad curvature increase, why does it happen specifically for AA? Do the authors have a reasonable justification?
4. Some minor changes would make the paper easier to read.
    1. For example, I suggest the authors move Figure 1 to Section 3 rather than Section 1. (I don’t see the reason to put Figure 1 in Section 1, whereas it is not mentioned in Section 1.)
    2. Similarly, Figure 2 can be moved to Section 5. While it is mentioned in Section 3, the readers would be more interested in the findings in Section 5 rather than the advantage of the authors’ design choice.


**Strengths And Weaknesses:**

# Strength
1. The authors provided ample experiments covering both breadth and depth of research questions regarding CO.
2. The experimental results are non-trivial enough to intrigue some other readers’ interests.
    1. In particular, the experiments covering the reason for the curvature increase (Section 5) contains some non-trivial finding regarding CO.

# Weakness
1. I’m not sure if the findings reflect some realistic scenarios and if the results explain the reason for CO in reality.
    1. In Section 4, The authors point out the networks’ preference for the injected features $\mathcal{V}$. Also, in Section 5, the authors point out that the interaction between the feature in the input $x$ and the injected feature $\mathbf{v}(y)$ is the cause of CO. However, where are those label-dependent features in the real world?
    2. I’m worried that the findings could be limited to the authors’ experiment setting. To explain, the existence of the injected features is crucial, and the findings might not generalize in real-world CO that does not involve such injected features.
    3. If the authors have some good way to justify this difference (between the experiment setting and the real-world scenario), such justification should appear in the paper.

---

> ### Author Response · Authors · 2023-05-31
> **Reply to reviewer z6NM (1/2)**
>
> We would like to thank the reviewer for their time and valuable feedback. We appreciate that they found our work provides “*ample experiments covering breadth and depth*” and that it can “*intrigue other reader’s interests*”.
>
> Below we address the main concerns of the reviewer:
>
> > I’m not sure if the findings reflect some realistic scenarios and if the results explain the reason for CO in reality.
>
> Although we understand the concerns of the reviewer, we disagree with this assessment. Deep learning is an empirical discipline, where many insights are gained from experimental studies, and as shown by prior work (Arpit et al., 2017; Ilyas et al., 2019; Shah et al., 2020; Ortiz-Jimenez et al., 2020a), manipulating data in controlled ways is a powerful tool to infer the structure of its features. In this light, we are confident our work provides robust and comprehensive evidence that features are likely the primary catalyst of CO across different datasets:
>
> - We provide many ablations with other types of injected features (e.g. random directions, other norms, etc.) in Appendix D and found they also lead to CO.
> - In the old Appendix G (and now in the main text) we show that the curvature increase observed in our injected datasets matches the pattern in naturally occurring CO settings.
> - In Section 7, we show the opposite effect and show that removing some features (in this case high frequency components) can sometimes prevent CO. This is another indication that data regulates CO.
> - Both N-FGSM and GradAlign were designed to prevent naturally occurring CO, but we show they can also prevent CO on our injected datasets. This suggests both types of CO are the same phenomenon.
> - In Appendix B we prove theoretically there exist many learning problems in which a robust classifier requires leveraging additional non-linear features on top of the simple (linear) ones used for the clean solution.
>
> **Why don’t we identify the features that cause CO in natural scenarios?**
>
> The key element of our controlled experiments is that we can synthetically modify the robustness of the injected features, and thus control the onset of CO. However, to perform similar experiments on unmodified datasets would require identifying the non-robust features. This is far from a trivial problem that would require new algorithmic advances beyond what is possible with our current deep learning tools.  In fact, doing so would be virtually equivalent to solving robustness, as if one can identify and manipulate the non-robust features of natural datasets, then one could also remove them entirely.
>
> Finally, we would like to highlight that very important papers in the field have also relied on data interventions. In particular, the influential (Ilyas et al. 2019) also uses data interventions to boldly claim that “adversarial examples are weak features”. In our opinion, the burden of evidence in our work is comparable to the one in this article.
>
> That being said, we agree with the reviewer that the original version of our manuscript did not discuss deep enough the potential generality of our claims. In this regard, we have modified the end of Section 6 with the hope to engage more deeply in such discussion. We have also moved new Figure 5 from the Appendix to the main body of text to highlight more strongly the parallels between the dynamics of our induced CO and naturally occurring CO.

---

> > ### Author Response · Authors · 2023-05-31
> > **Reply to reviewer z6NM (2/2)**
> >
> > > [...] after the model learns with the meaningless FGSM perturbations, does the model just lose robustness, or it starts focusing on the meaningless FGSM perturbation?
> >
> > This is a very interesting question which has been studied by prior work. In particular, Wong et al. (2020), Andriushchenko and Flammarion (2020) and Kim et al. (2021) all showed that after CO, FGSM does no longer act as an effective adversarial attack, and as such adding its discovered solutions does not have a significant effect in accuracy. Furthermore, de Jorge et al. (2022) showed that after CO the diversity of the solutions of FGSM, as measured by the effective rank of a collection of these perturbations, increases. After CO, the FGSM perturbations are not consistent across epochs and do not seem to meaningfully converge to any particular direction. These observations, alongside our results in Section 5 highlight that after CO, FGSM does not point towards any meaningful perturbation, and its landing direction is probably arbitrary.
> >
> > Using the FGSM perturbations to probe the features used by the model to classify is futile, and using pixel attribution methods would probably be as well. Not only would the curvature increase after CO probably render any saliency method ineffective, but also the features in $\mathcal{V}$ are visible all over the image and not localized. In this regard, a technique like GradCAM which is designed to identify important local semantic features in an image would be completely useless. However, thanks to the fact that we control which features are present in our dataset, we can use the technique of Figure 2 and see that after CO the model does mainly focus on the features in $\mathcal{V}$ and completely ignores the CIFAR-10 features.
> >
> > > Do the authors have some experimental support that the curvature increase (by AutoAttack) actually hindered PGD optimization, or is there any other reason for this?
> >
> > We would like to clarify that all the experiments of Section 7 were trained using FGSM-AT and that PGD-10 and AutoAttack were only used to evaluate the models. In this regard, only FGSM-AT can be responsible for the curvature increase in these experiments, as neither PGD-10 nor AutoAttack were used during training.
> >
> > In Section 7, we highlight the interesting observation that when CO happened in our injected ImageNet-100 experiments we were only able to attack those models using AutoAttack (which uses a combination of stronger white-box attacks and a black box attack insensitive to changes in local curvature). Attacking with PGD-10 in those cases behaved the same way as attacking with FGSM, and for this reason we claim that our induced CO in ImageNet-100 also hindered the optimization landscape for PGD.
> >
> > > Some minor changes would make the paper easier to read.
> >
> > We thank the reviewer for these suggestions which we have duly implemented in the new version of our manuscript.

---

### Review · Reviewer_KwEe · 2023-05-30

**Summary Of Contributions:**

This paper empirically studies the causes of catastrophic overfitting (CO) in FGSM adversarial training (AT). In particular, this paper proposes a novel data intervention technique to induce CO. To this end, a set of linearly separable data points are added to the original data. By controlling the magnitude of these easy-to-learn features, the paper aims to force CO. Conducting extensive exploratory experiments on various datasets and models, the paper concludes that CO happens when the network prefers easy non-robust features over hard-to-learn robust ones. Using these observations, the paper makes interesting connections between the observed behaviour and well-known phenomena such as shortcut learning and curvature increase in deep neural networks. A preliminary approach to using these observations in mitigating CO is also proposed. Motivated by the reliance of DNNs on high-frequency features, the authors suggest using a simple low-pass version of CIFAR-10 to prevent CO. However, this method is shown to be effective in $\epsilon=8/255$ perturbation norm.

**Audience:**

Yes

**Broader Impact Concerns:**

The current submission needs to provide a broader impact section. It would be nice if the authors could provide such a statement for the general audience.

**Claims And Evidence:**

Yes

**Requested Changes:**

As mentioned above, I believe that even if the authors cannot provide a viable working solution based on the observations made, it would be beneficial to expand on Sec. 7. In other words, I encourage them to discuss the possible ways that these empirical observations could be incorporated to mitigating CO in practice. This would make the paper more interesting.

**Strengths And Weaknesses:**

### Strengths:

- The paper thoroughly investigates CO in FGSM-AT. The design of the data intervention through linearly separable perturbations is innovative and provides flexibility to verify CO in different settings empirically. Furthermore, the connections made to shortcut learning and heavy regularisation through GradAlign put the paper well within the literature.

- The paper is well-structured and easy to read. It takes the reader through each step of the empirical investigation and provides sufficient reasoning for the observations made.

- The experiments are conducted over multiple datasets (CIFAR-10, CIFAR-100, SVHN, TinyImageNet, and ImageNet-100) and models (ResNet and WideResNet). This would make the results likely to be reproducible.

### Weaknesses:

- Currently, the paper only provides interesting observations into CO. The paper needs to delve deeper into the consequences of these observations and how we can exploit them to mitigate CO. Currently, it only investigates existing options, such as GradAlign, or presents a low-pass filtration technique that would only work for small perturbation magnitudes.

---

> ### Author Response · Authors · 2023-05-31
> **Reply to Reviewer KwEe**
>
> We would like to thank the reviewer for their time and kind words. We appreciate that they found our work ”*thorough*”, “*well-positioned with respect to the literature*”, “*well-structured*”, “*sufficiently reasoned*” and “*reproducible*”.
>
> Below we address the main requests of the reviewer:
>
> >  I encourage [the authors] to discuss the possible ways that these empirical observations could be incorporated to mitigating CO in practice.
>
> We appreciate the reviewer’s suggestion and understand their position. However, we would like to highlight that the main objective of this work is not providing a new methodology, but explaining a strange phenomenon that is poorly understood. There already exist efficient methods to prevent CO in fast-AT. Specifically, N-FGSM (de Jorge et al., 2022) has managed to close the computational gap between FGSM-AT (which suffers from CO) and CO-safe fast-AT methods, as it provides a cheap solution to the problem by augmenting the data with random noise with the same cost as FGSM-AT. In this regard, it is highly unlikely that one can find a simpler solution to CO than N-FGSM based on our insights. Especially, as any such solution would involve identifying non-robust and robust features which is still an open problem in the field.
>
> That being said, the root cause of CO is still not understood, which makes, in our opinion, the analysis of this paper more timely than a new method. The deep learning literature at large is full of  many influential works of this nature, e.g., Zhang et al. (2017), Ilyas et al. (2019), Frankle & Carbin (2019), etc. These works did not provide any new method, nor solve any particular problem, and instead explained and analyzed complex phenomena of neural networks and became valuable contributions to their corresponding fields. For this reason, we politely disagree with the reviewer as we believe that not presenting a new method to solve CO does not weaken the relevance of our work.
>
> - Zhang et al. Understanding deep learning requires rethinking generalization. ICLR 2017
> - Ilyas et al. Adversarial examples are not bugs, they are features. NeurIPS 2019
> - Frankle & Carbin. The lottery ticket hypothesis: Finding sparse, trainable neural networks. ICLR 2019
>
> > Broader impact
>
> We have added a new Broader Impact statement where we discussed the social impact of our work in the new version of the paper.

---

### Author Response · Authors · 2023-05-31
**Manuscript revision**

We thank the reviewers for their comments and their valuable feedback. We have answered the main concerns of each reviewer directly as replies, but we have also made some changes to the manuscript we would like to highlight:

- We have introduced a new discussion about the generality of our claims at the end of Section 6.
- We have moved our experiments analyzing the curvature of naturally occurring CO from the Appendix to the main text (now in Section 5).
- We have placed Figure 1 and Figure 4 in better positions.
- We have fixed the comment about the directionality of the FGSM perturbations prior to CO of Section 5.1.
- We have fixed the typo in Eq. (3).
- New Broader impact statement.

We hope these changes can satisfy the requests of the reviewers and we remain open to any new requests they may have to improve our text.

---

### Decision · Action_Editors · 2023-07-14

**Recommendation:** Accept as is

**Comment:**

All three reviewers suggest accepting the paper (1 x accept, 2 x leaning accept), thus, there is a clear consensus.
The authors put a lot of effort into addressing comments from reviewers. The updated version of the paper reads very well and, in my opinion, it does not require further corrections. As a result, I propose to accept the paper as is.

**Audience:**

The paper presents an interesting insight into the problem of catastrophic overfitting. As highlighted by one of the reviewers (z6NM) in their official recommendation: "The paper provides well-supported claims and evidence, and I believe that the findings are nontrivial enough to intrigue other TMLR audience". I fully agree with this statement.

**Claims And Evidence:**

The paper explores the root cause of the problem of *catastrophic overfitting* (CO), a phenomenon in which a neural network reaches a breaking point beyond which it loses robustness. The claims of the paper are the following:
(1) CO can be induced by injecting features that are strong for standard classification but insufficient for robust classification.
(2) CO is connected to the network’s preference for certain features in a dataset.
(3) There is a mechanistic explanation of CO.

The paper first provides an analysis of induced CO. By looking into injected features and geometry of CO, the authors provide an interesting perspective supporting their claim (1).
Next, the authors summarize their observations in a mechanistic explanation of CO. This part is a validation of their claim (3).
Eventually, in a series of experiments, they present empirical evidence for their claim (2).